# Bifrost-1: Bridging Multimodal LLMs and Diffusion Models with Patch-level CLIP Latents

**Han Lin**[1] **Jaemin Cho**[1] **Amir Zadeh**[2] **Chuan Li**[2] **Mohit Bansal**[1]

[1]UNC Chapel Hill    [2]Lambda

## Abstract

There is growing interest in integrating high-fidelity visual synthesis capabilities into large language models (LLMs) without compromising their strong reasoning capabilities. Existing methods that directly train LLMs or bridge LLMs and diffusion models usually suffer from costly training since the backbone LLMs have not seen image representations during pretraining. We present BIFROST-1, a unified framework that bridges pretrained multimodal LLMs (MLLMs) and diffusion models using patch-level CLIP image embeddings as latent variables, which are natively aligned with the MLLM's CLIP visual encoder. These patch-level image embeddings are integrated into the diffusion model with a lightweight adaptation of its ControlNet. To retain the original multimodal reasoning capabilities of MLLMs, we equip the MLLM with a visual generation branch initialized from the original MLLM parameters when predicting the patch-level image embeddings. By seamlessly integrating pretrained MLLMs and diffusion models with patch-level CLIP latents, our framework enables high-fidelity controllable image generation with significant training efficiency. Our experiments demonstrate that BIFROST-1 achieves comparable or better performance than previous methods in terms of visual fidelity and multimodal understanding, with substantially lower compute during training. We also provide comprehensive ablation studies showing the effectiveness of our design choices. Project page: https://bifrost-1.github.io.

## 1 Introduction

Humans can naturally process and generate multimodal information such as language, vision, and sound. This ability stems from our integrated cognitive system capable of reasoning, organization, and expression. Inspired by this, building a unified AI system that supports both multimodal understanding and generation has become an active area of research [48, 10, 80, 62, 73, 68]. On the one hand, large language models (LLMs) have achieved impressive results across diverse tasks, including natural language understanding [17, 40, 6, 54], mathematical reasoning [33, 58], and code generation [9, 46], demonstrating strong planning and reasoning abilities [49, 26]. On the other hand, recent advances in image and video generation models have enabled high-fidelity visual synthesis [50, 74, 19, 47]. Therefore, effectively and efficiently integrating visual generation capabilities into LLMs, while preserving their reasoning capabilities, has become an active research direction in the pursuit of unified AI systems that bridges language understanding and visual synthesis.

Existing LLM-based image generation methods fall into two main paradigms. First, single-architecture approaches (see Fig. 1 (a)) employ an LLM/MLLM that takes concatenated image and text tokens as inputs. The model is then finetuned, either by updating all LLM parameters [12, 67, 42, 41, 62, 71, 10, 68] or via an auxiliary visual generation branch [44, 37, 59], to generate images directly. While this offers unified reasoning and native image synthesis, it typically incurs very expensive computational costs to achieve high-fidelity results. Moreover, without careful

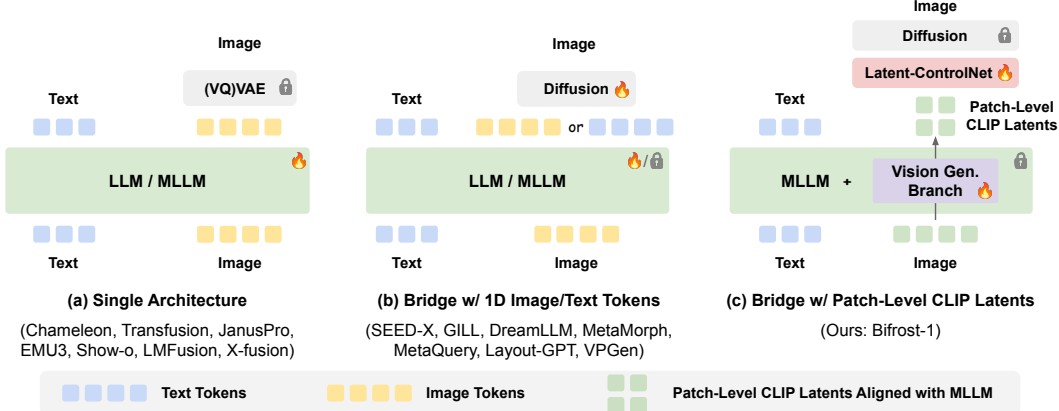

Figure 1: **Comparison of different approaches in using LLM for image generation**. (a) Single architecture handling both text and image tokens into the LLM/MLLM. (b) Bridging LLM and Diffusion model with a 1D sequence (image tokens or text tokens). (c) BIFROST-1 (ours), which bridges MLLM with diffusion models with 2D image tokens aligned with MLLM embeddings.

balancing of training data and objectives, such finetuning can risk degrading the LLM's original language reasoning capabilities [71], thereby undermining a primary goal of the integration.

Second, bridging-based approaches (see Fig. 1 (b)) use both pretrained diffusion models as well as LLMs, teaching an LLM to guide a diffusion model in generating the final image. The guidance takes the form of detailed text descriptions (e.g., object layouts, extended prompts) generated by the LLM [13, 20, 14], or continuous query tokens also produced by the LLM [31, 23, 63, 51, 1]. Although generally more training-efficient than single-architecture methods, these approaches have their own limitations. Text-based representations can be too sparse to capture fine-grained details of complex scenes, while learning to generate effective 1D continuous query tokens often requires a large connector and end-to-end training, and may not sufficiently convey detailed 2D spatial information.

To address these limitations, we introduce **BIFROST-1** (Sec. 3). As illustrated in Fig. 1(c), BIFROST-1 uniquely bridges a pretrained multimodal LLM (MLLM) [5] and a diffusion model [32]. Our core innovation lies in using 2D patch-level CLIP [53] latents as the communicative medium. These latents are image embeddings that are natively aligned with the MLLM's own CLIP visual encoder, enabling the MLLM to generate rich and precise spatial guidance for image synthesis. These patch-level latents are integrated into the diffusion model via a lightweight adaptation of its ControlNet [78]. To retain the original multimodal reasoning capabilities of MLLMs, we equip the MLLMs with a visual generation branch initialized from the original MLLM parameters when predicting the patch-level image embeddings. By seamlessly integrating pretrained MLLMs and diffusion models with patch-level CLIP latents, our framework enables high-fidelity controllable image generation with substantial training efficiency. In addition, by fully retaining the original parameters of the MLLM backbone, BIFROST-1 does not suffer from performance degradation of multimodal reasoning capabilities observed in some previous works.

We empirically demonstrate the effectiveness of BIFROST-1 through a series of experiments. First, in Sec. 5.1, we compare different architectures for bridging LLMs and diffusion models, showing that patch-level CLIP latents train more efficiently than the controlled variants with alternative methods. Next, in Sec. 5.2, we present the effective scaling of the number of CLIP latent tokens to improve image reconstruction quality. We also qualitatively demonstrate that the image reconstruction quality of BIFROST-1 is competitive or superior to strong baselines with a magnitude higher training computation cost. In Sec. 5.3, we compare BIFROST-1 with state-of-the-art methods on multimodal understanding and generation tasks, showcasing its competitive performance and the preservation of the MLLM backbone's reasoning capabilities. Finally, in Sec. 5.4, we experiment with varying number of MLLM decoding steps, and observe that our method is robust as long as the number of decoding steps is larger than 8. In a nutshell, our contribution can be summarized as follows:

- We propose a novel framework (BIFROST-1) for unified multimodal generation and understanding by bridging pretrained MLLM and diffusion models with fine-grained patch-level CLIP latents.

- By using CLIP latents that are natively aligned with the MLLM, we achieve significantly lower training costs compared to recent works with a single architecture or those that bridge using 1D latents or query tokens.
- We show qualitatively and quantitatively that BIFROST-1 achieves comparable performance with previous SoTAs on image reconstruction quality and text-to-image generation benchmarks, while well-preserving MLLM's original multimodal understanding ability.

## 2  Related Work

**Unified multimodal LLM architectures for visual generation.** The success of large language models (LLMs) such as T5 [54] and GPT-3 [6] has driven efforts toward extending language modeling paradigms to multimodal settings. Early works like VL-T5 [11], SimVLM [69], and Flamingo [3] adapt pretrained LLMs for visual understanding by framing perception tasks as text generation. More recent models also incorporate image generation capabilities into LLMs using either autoregressive objectives [12, 67, 42, 41, 62, 71, 10, 68, 59, 44] or denoising diffusion objective [73, 80], as described in Fig. 1 (a). While these approaches enable flexible, end-to-end training, they often inherit their initialization from text-only LMs and require significant resources to learn high-quality image generation from scratch. This limitation has motivated a complementary line of research: bridging pretrained LLMs with diffusion models, as discussed next below.

**Bridging LLM and diffusion models for visual generation.** As described in Fig. 1 (b), several methods decompose the generation process by using LLMs to produce detailed intermediate textual representations, such as object layouts or detailed scene descriptions, which guide a pretrained diffusion model [13, 20, 14]. However, complex visual scenes with rich object interactions are often hard to describe purely with text. An alternative strategy is to replace intermediate text with continuous visual representations to guide a diffusion model [31, 23, 63, 51]. For example, Tong et al. [63] starts with an LLM and conduct large-scale training to equip it with a SigLIP [77] visual encoder. The image tokens generated autoregressively by the LLM are supervised using a cosine similarity loss with ground-truth SigLIP embeddings. These predicted image tokens are then injected into diffusion models via cross-attention. However, teaching LLMs to generate visual representations still requires more text-image annotations and computational resources (see Table 1), and it is also prone to degraded text-generation quality unless the model is carefully balanced and trained on high-quality text-text and vision-text data, along with extensive compute. Hence, we explore the use of multimodal LLMs (MLLMs) that are already strongly aligned with CLIP embeddings encoding spatial information, by teaching them to generate 2D CLIP embeddings to guide diffusion models through latent ControlNet. As shown in Sec. 5.1, we find our method of bridging MLLM and Diffusion models with 2D CLIP embeddings and latent ControlNet is significantly more data-efficient in broad training budget scenarios compared to the above baselines.

## 3  BIFROST-1

We introduce **BIFROST-1**, a novel unified multimodal framework that effectively bridges pretrained MLLMs and diffusion models with patch-level CLIP latents which are natively aligned with the MLLM. This enables MLLMs to generate rich and precise spatial guidance for image synthesis while achieving significant training efficiency. In the following, we present preliminaries about MLLMs and ControlNet (Sec. 3.1), how BIFROST-1 bridges MLLMs and diffusion models via patch-level CLIP latents (Sec. 3.2), and training and inference strategy (Sec. 3.3).

### 3.1  Preliminaries

**Visual encoding and decoding with MLLMs.** For image understanding tasks, given an RGB image $x \in \mathbb{R}^{3 \times H \times W}$, recent MLLMs commonly employ visual encoders that convert the image into a set of $d$-dimensional continuous (e.g., CLIP [53]) or discrete (e.g., VQVAE [64]) visual embeddings: $z = \mathcal{E}(x) \in \mathbb{R}^{d \times H' \times W'}$, where $H' < H$ and $W' < W$ due to spatial downsampling [5, 81, 2, 15]. These visual embeddings are aligned with the LLMs' embedding space via learned projection layers. For image generation tasks, to enable the MLLM to produce images, the visual embeddings generated by the LLM backbone are passed to a visual generation decoder $\mathcal{D}$ to render the final image: $x = \mathcal{D}(z)$. Typically, VQVAEs or diffusion models are employed for the decoder, whose visual

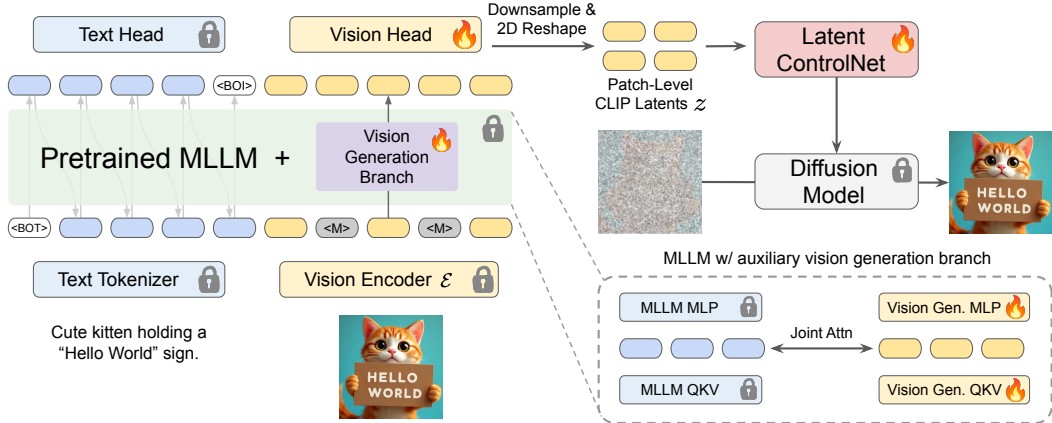

Figure 2: **Overview of BIFROST-1**. BIFROST-1 equips the backbone MLLM with a visual generation branch, which is a trainable copy of a pretrained MLLM parameters (i.e., QKV, MLP, normalization layers) and a newly added vision head (i.e., a linear layer). The visual generation branch outputs patch-level CLIP latents, which are then downsampled and reshaped into 2D (HxW), provided to latent ControlNet, and finally guiding image generation of a pretrained diffusion model. During training, a portion of the image patches is randomly replaced with learnable mask tokens <M>. During inference, we start with fully masked image tokens and autoregressively predict them.

representation is different from CLIP embeddings; thus, learning to align MLLMs and decoders usually requires significant computation. In this work, we extend unCLIP [55] that employs a model to convert a single (pooled) CLIP text embedding to a single (pooled) CLIP image embedding by teaching MLLM to generate patch-level CLIP latents which are natively aligned with the visual encoder $\mathcal{E}$ of the pretrained MLLM. We find this significantly facilitates image generation training.

**ControlNets.** ControlNet [79] is designed to add spatial controls with image guidance inputs (*e.g.*, depth, sketch, segmentation maps, *etc.*) to diffusion models. Specifically, given a pretrained diffusion model $\mathcal{F}_{\theta}$, ControlNet adopts a similar architecture $\mathcal{F}_{\theta'}$, whose parameters are initialized from $\theta$ (i.e., 'trainable copy'). ControlNet takes the diffusion timestep $t$, text prompt $c_{\text{text}}$, control image $c_f$ (*e.g.*, depth map), and the noisy latents $z_t$ as inputs, and outputs features guide the backbone diffusion $\mathcal{F}_{\theta}$ for the final image generation. In this work, we propose latent ControlNet that uses patch-level CLIP image embeddings instead of a control image to guide diffusion models.

## 3.2 Bridging MLLMs and Diffusion Models with Patch-Level CLIP Latents

**BIFROST-1 design summary.** We have two main goals: (1) to preserve the multimodal understanding capability of the MLLM, while (2) efficiently teaching MLLM to generate latent tokens to guide diffusion models. As illustrated in Fig. 2, BIFROST-1 achieves these goals by having a new visual generation branch initialized from the original MLLM parameters and using patch-level CLIP image embeddings as latent bridge tokens.

**Learning to unmask image patch embeddings with a visual generation branch.** To teach an MLLM image generation, we first encode the images using the MLLM's native visual encoder to get patch-level image embeddings and concatenate them with text tokens. Following MAR [36], we replace parts of the input image embeddings with a learnable mask token (<M>) and let the MLLM to predict the masked image embeddings. The mask ratio is randomly sampled from a truncated normal distribution with a mean of 1.0 and a standard deviation of 0.25, constrained to the range [0.7, 1.0]. For this image embedding prediction task, we introduce a visual generation branch (Fig. 2 left), whose parameters are initialized from MLLM parameters (i.e., attention QKV projections, MLP projection layers, and normalization layers) following LMFusion [59]. Just like text head, which is a linear layer toward text embedding space, we use a simple linear layer as a vision generation head. By reusing majority of parameters from the pretrained MLLM and randomly initializing only a single linear layer as the vision head, we avoid the costly process of realigning image embeddings.

**Attention across modalities.** In Fig. 3, we illustrate how different image and text input tokens can attend to each other. In summary, we apply causal masking for text, and we apply full attention for image tokens. All the previous modalities are fully visible to future modality tokens.

**Latent ControlNet.** To effectively guide diffusion models with patch-level CLIP image embeddings, we create latent ControlNet (Fig. 2 top right), by modifying the original ControlNet architecture from a backbone image diffusion model (*e.g.*, FLUX.1-dev [32]) with the following two changes:

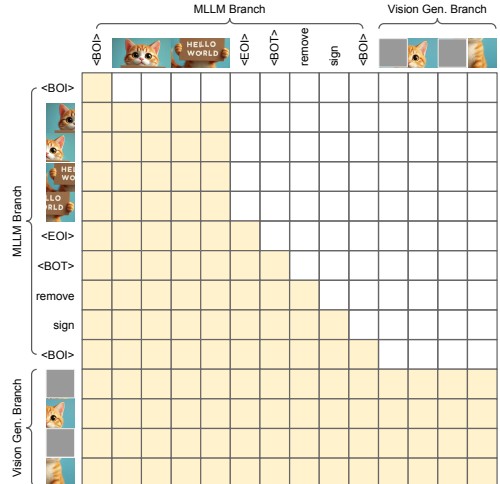

- We replace the input linear projection layer ($\mathbb{R}^{3 \times d} \to \mathbb{R}^{d' \times d}$), where $d$ is the embedding dimension of the backbone image diffusion model, and $d'$ is the embedding dimension of the MLLM.
- To further reduce the number of visual tokens generated by the MLLM, we introduce a lightweight downsampling 2D convolution block to reduce the spatial resolution of the visual embeddings by a factor of 2 before passing to latent ControlNet.

Figure 3: Attention mask. X-axis is input and Y-axis is output.

During training, we update only the newly added input linear projection, the 2D downsampling convolution blocks, and the ControlNet for 4 MM-DiT blocks and 1 Single-DiT block, compared to the full FLUX.1-dev model which contains 19 MM-DiT blocks and 38 Single-DiT blocks in total.

### 3.3 Training and Inference

**Decoupled training.** The MLLM visual prediction branch and latent ControlNet can be jointly trained or separately. We adopt the second, a decoupled training strategy for the following two reasons: (1) Since we only train small parameters of latent ControlNet, it converges much faster than the MLLM's vision branch. In initial experiments, we find that decoupled training allows us to allocate more compute to the MLLM's vision branch, improving overall performance. (2) Since recent diffusion models have large parameters (*e.g.*, FLUX.1-dev has 12B parameters), end-to-end training significantly increases memory requirements. For the MLLM visual prediction branch, we use the mean squared error (MSE) loss for image patch embedding prediction. For the latent ControlNet, we use the original flow-matching loss used in FLUX ControlNet.

**Inference of image patch embeddings tokens via masked autoregression.** During inference, the MLLM is given a text prompt along with fully masked CLIP image embeddings. Following MAR [36], we first randomly sample a generation order of image patches, then iteratively denoise them into clean CLIP image embeddings. Once all image patches are obtained, we give them latent ControlNet to guide the diffusion model.

## 4 Experiment Setup

**Latent ControlNet implementation details.** For the latent ControlNet design, we build upon the official FLUX.1-dev ControlNet training implementation from the Hugging Face Diffusers library.[1] Specifically, we train a ControlNet consisting of 4 MM-DiT [19] (DoubleStream) blocks and 1 Single-DiT [52] (SingleStream) block (i.e., `num_double_layers=4, num_single_layers=1`) with a total batch size of 48. All other training hyperparameters, including learning rate, Adam optimizer, and weight decay, are kept identical to the original codebase without any tuning. During latent ControlNet inference, we also retain all default hyperparameters unchanged (e.g., `num_inference_steps=28`, `controlnet_conditioning_scale=0.7`, `guidance_scale=3.5`).

**Baselines.** For ImageNet experiments, we compare BIFROST-1 with two main variants, using the same model initialization and computational budget: (1) the first variant replaces the MLLM-aligned

---

[1]https://github.com/huggingface/diffusers/tree/main/examples/controlnet

Table 1: Comparison of different architectures for bridging LLMs and diffusion models in image generation on ImageNet 256×256.

| Method | FID↓ | sFID↓ | IS↑ |
|---|---|---|---|
| Backbone diffusion model | | | |
| FLUX.1-dev [32] | 51.20 | 224.82 | **157.46** |
| 2D Learnable query tokens instead of CLIP latent | | | |
| MLLM + **2D Learnable Query Tokens** + Latent ControlNet + FLUX.1-dev | 118.69 | 129.14 | 9.15 |
| No pre-aligned visual encoder instead of pre-aligned visual encoder | | | |
| MLLM + **SigLIP** + Latent ControlNet + FLUX.1-dev | 274.16 | 304.94 | 2.69 |
| VAE instead of CLIP latent | | | |
| MLLM + **FLUX VAE** + Latent ControlNet + FLUX.1-dev | 284.51 | 361.45 | 1.11 |
| Cross-attention instead of Latent ControlNet | | | |
| MLLM + Patch-level CLIP Latent + **cross-attention** + FLUX.1-dev | 76.32 | 208.11 | 26.35 |
| BIFROST-1 (Ours) | | | |
| MLLM + **Patch-level CLIP Latent** + Latent ControlNet + FLUX.1-dev | **25.77** | **53.67** | 98.57 |

CLIP-based embeddings with non-MLLM-aligned FLUX VAE features for visual generation, and (2) the second variant uses 2D query tokens (extended from MetaQuery [51]) instead of masked autoregression for visual generation. For SoTA comparison experiments, we compare our method with baselines for unified multimodal understanding and generation including DreamLLM [18], Chameleon [62], Show-o [73], VILA-U [72], EMU3 [68], MetaMorph [63], MetaQuery [51], Token-Flow [24], Transfusion [80], LMFusion [59], Janus [71], JanusFlow [43], and JanusPro [10].

**Training details.** We train BIFROST-1 on ImageNet [16] for the experiments in Sec. 5.1 and Sec. 5.2. Specifically, the latent ControlNet and BIFROST-1 MLLM in Sec. 5.1 are trained for 2 epochs and 16 epochs respectively, and the latent ControlNet in Sec. 5.2 is trained for only 1 epoch (∼25M training steps). For SoTA comparison experiments in Sec. 5.3, we measure the training cost by the total number of training images passed to the model, calculated as the product of the number of training steps and the batch size per step. Our models with Qwen2.5-VL 3B/7B are trained on 9M and 62M images respectively from the BLIP3-o [8] training dataset.[2] All experiments on ImageNet (Sec. 5.1 and Sec. 5.2) are trained on a single GH200 GPU, and the SoTA comparison experiments in Sec. 5.3 are trained on 16 GB200 GPUs.

**Evaluation datasets and metrics.** For ImageNet, following previous works [36, 56], we evaluate our model on Fréchet Inception Distance (FID) [27], sFID [45], and Inception Score (IS) [57]. Then for open prompt evaluation, we follow previous works [10, 51, 63, 59] and report FID scores on MJHQ-30K [35] and 30k randomly sampled images from MSCOCO [38] validation set for visual aesthetic quality, and GenEval [25] and DPG-Bench [28] for prompt alignment, respectively. We use the original prompts from all these four evaluation datasets directly without any prompt rewriting.

## 5 Results and Discussion

We empirically demonstrate the effectiveness of BIFROST-1 in various experiments. Specifically, we compare different architectures for bridging LLMs and diffusion models (Sec. 5.1), training efficiency of different numbers of CLIP latent tokens for image reconstruction (Sec. 5.2), BIFROST-1 with SoTAs on multimodal understanding and generation tasks (Sec. 5.3), and experiments with different MLLM decoding steps (Sec. 5.4).

### 5.1 Comparison of Design Choices for Bridging LLMs and Diffusion Models

We compare different choices for bridging LLMs and diffusion models in terms of image generation task on ImageNet 256x256. We generate 10K images with classes randomly sampled from 1k categories and compute visual quality metrics. In all settings, the MLLM visual generation branch and the Latent ControlNet are trained for 16 and 2 epochs, respectively.

---

[2]https://huggingface.co/datasets/BLIP3o/BLIP3o-Pretrain-Long-Caption

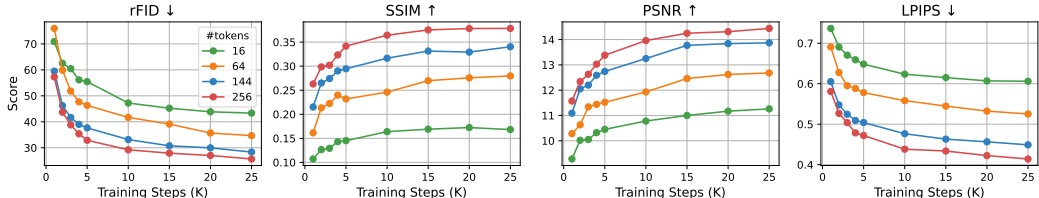

Figure 4: Image reconstruction scores with different numbers of 2D CLIP latent tokens used within BIFROST-1 on ImageNet for around one training epoch (∼26K steps). Results indicate that using more tokens achieves better data efficiency.

**BIFROST-1 vs. Backbone diffusion model.** In the first and last rows of Table 1, we compare BIFROST-1 and its backbone diffusion model (FLUX.1-dev [32]). We find that BIFROST-1 improves FID and sFID, while it hurts IS from the original backbone. The improvement over the baseline mainly comes from adding a few trainable ControlNet blocks to the backbone diffusion model, which enables better adaptation to the data distribution.

**Patch-level CLIP latent vs. 2D learnable query tokens.** MetaQuery [51] is a recent method that bridges LLMs and diffusion models by learning a connector (24-layer transformer encoder) that projects a frozen LLM's hidden representations on a finite number of learnable query tokens, where the connector outputs guide a diffusion model via cross attentions. Inspired by this, we implement a 2D version of MetaQuery and compare it with BIFROST-1. To fairly compare methods with similar additional parameters, instead of learning a heavy connector module, we directly reshape the LLM representations of learned query tokens as inputs to the latent ControlNet. As shown in Table 1, 2D learnable query tokens hurt performance in all 4 metrics, indicating that learning to align representations between MLLM and diffusion from scratch requires much more computation than our patch-level CLIP latent.

**Patch-level CLIP latents vs. VAE latents.** Here, we compare BIFROST-1 with a variant that replaces CLIP latents (that are natively aligned with MLLM) with VAE latents (that are not originally aligned with MLLMs). Specifically, we substitute the CLIP visual encoder with the FLUX VAE encoder, and replace our visual decoder (i.e., Latent ControlNet + FLUX diffusion model) with the FLUX VAE decoder. Linear projection layers are applied to align the dimensions of the VAE-encoded features with the feature dimension of the MLLM backbone. All other components of our framework and the training strategy are kept the same to ensure a fair comparison. As shown in the 3rd row of Table 1, using VAE features significantly slows down learning. Additionally, this highlights that by leveraging a pretrained diffusion model as the image renderer, BIFROST-1 effectively reduces the burden on the MLLM side to directly generate high-quality images.

**MLLM's native visual encoder vs. non-aligned external visual encoder.** In addition, we also compare using the latents from MLLM's native visual encoder (*i.e.*, CLIP) with an external visual encoder (*i.e.*, SigLIP, as used in MetaMorph [63]). As we can see in the 4th row of Table 1, although using SigLIP achieves better image generation quality than using VAE, it is still significantly worse than using the MLLM's native visual encoder. This indicates the training efficiency of adopting MLLM's native visual encoder compared with non-aligned external visual encoders.

**Diffusion model guidance strategy: Latent ControlNet vs. Cross-attention** Our method directly injects 2D image tokens generated by the MLLM into the DiT via latent ControlNet. By adding 2D ControlNet latents on top of the noisy DiT latents, our method can enforce spatial structures more explicitly and effectively. In contrast, several previous works [63, 51, 8] use cross-attention to condition on image tokens. We conducted an additional experiment comparing these two conditioning strategies. For a fair comparison, we unfreeze all parameters in DiT when conditioning it via cross-attention and use 64 MLLM-generated image tokens for both methods. As shown in the 5th row of Table 1, our latent ControlNet achieves better image generation quality (FID, sFID, IS), demonstrating its superior efficiency and effectiveness.

### 5.2 Image Reconstruction with Patch-level CLIP Latent

In the following, we demonstrate the effectiveness of CLIP Latent-based image representation in image reconstruction experiments. Concretely, we experiment with scaling the number of latent tokens and compare BIFROST-1 with SoTA models.

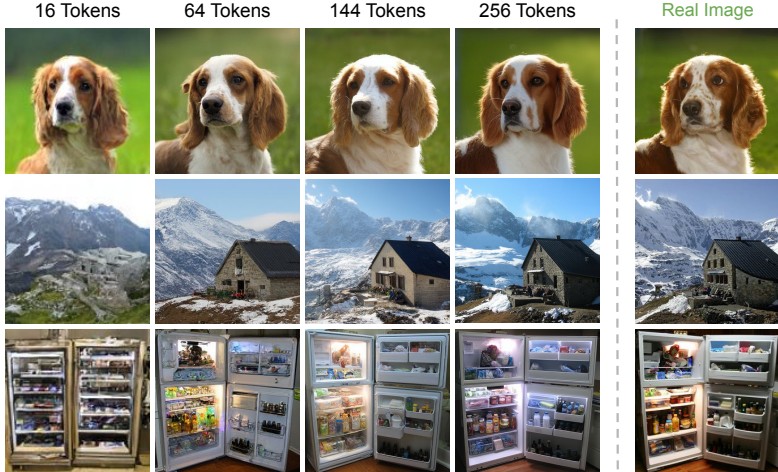

16 Tokens  64 Tokens  144 Tokens  256 Tokens  Real Image

Figure 5: Visual samples for image reconstruction with different numbers of patch-level CLIP tokens generated from MLLM. The Latent ControlNet models with varying numbers of tokens are trained for only 1 epoch on the ImageNet training split.

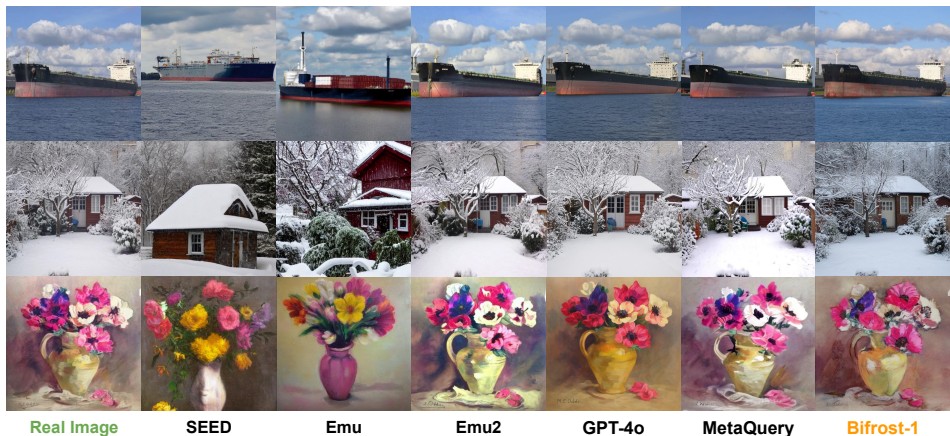

Real Image  SEED  Emu  Emu2  GPT-4o  MetaQuery  Bifrost-1

Figure 6:  Image reconstruction results.

**Scaling the number of CLIP latent tokens.** In Fig. 4 we show that image reconstruction quality scales well with the number of patch-level CLIP latent tokens input to the Latent ControlNet. With only 1 epoch of training (∼26K steps) on the ImageNet dataset, images represented by 256 (=14×14) CLIP latent tokens not only achieve higher reconstruction accuracy, measured by rFID, SSIM, PSNR, and LPIPS, but also converge faster compared to those using fewer CLIP latent tokens. In Fig. 5, we show qualitative examples of images generated with different numbers of CLIP latent tokens.

**Comparison with SoTA methods.** In Fig. 6, we qualitatively compare the image reconstruction quality of our latent ControlNet with various unified models, including SEED [22], EMU [61], EMU2 [60], GPT-4o [29], and MetaQuery [51]. Specifically, we first encode the images using the visual encoder in Qwen2.5-VL, then use the latent ControlNet to reconstruct the images based solely on this visual information, without providing any text prompt as additional guidance. Trained exclusively on the ImageNet dataset for 3 epochs without exposure to any other open-world images, the reconstructions from the BIFROST-1 latent ControlNet achieve quality that is competitive with or superior to strong baselines such as GPT-4o and MetaQuery, demonstrating the efficiency and robustness of our Latent ControlNet design.

## 5.3 Comparison with SoTA Models

**BIFROST-1 fully inherits strong visual understanding capabilities of backbone MLLM.** As BIFROST-1 keeps the original parameters in the MLLM completely frozen, the trainable vision branch is agnostic to the choice of MLLM. This allows BIFROST-1 to fully inherit the visual understanding

Table 2: Comparison with state-of-the-art methods on multimodal understanding benchmarks.

| Method | Base (M)LLM | MME-P↑ | MMB↑ | SEED↑ | MMMU↑ | MM-Vet↑ |
|---|---|---|---|---|---|---|
| DreamLLM [18] | Vicuna 7B | - | - | - | - | 36.6 |
| Chameleon [62] | From Scratch 7B | - | - | - | 22.4 | 8.3 |
| Show-o-512 [73] | Phi-1.5 1.3B | 1097.2 | - | - | 26.7 | - |
| VILA-U [72] | LLaMA-2 7B | 1401.8 | - | 59.0 | - | 33.5 |
| EMU3 [68] | From Scratch 7B | - | 58.5 | 68.2 | 31.6 | 37.2 |
| MetaMorph [63] | LLaMA-3 8B | - | 75.2 | 71.8 | - | - |
| MetaQuery-L [51] | Qwen2.5-VL 3B | 1574.3 | 78.6 | 73.8 | 53.1 | 63.2 |
| MetaQuery-XL [51] | Qwen2.5-VL 7B | 1685.2 | 83.5 | 76.9 | 58.6 | 66.6 |
| TokenFlow-XL [24] | Qwen-2.5 14B | 1551.1 | 76.8 | 72.6 | 43.2 | 48.2 |
| Transfusion [80] | From Scratch 7B | - | - | - | - | - |
| LMFusion [59] | LLaVA-Next 8B | 1603.7 | 72.1 | 72.5 | 41.7 | - |
| Janus [71] | DeepSeek-LLM 1.5B | 1338.0 | 69.4 | 63.7 | 30.5 | 34.3 |
| JanusFlow [43] | DeepSeek-LLM 1.5B | 1333.1 | 74.9 | 70.5 | 29.3 | 30.9 |
| JanusPro-1B [10] | DeepSeek-LLM 1.5B | 1444.0 | 75.5 | 68.3 | 36.3 | 39.8 |
| JanusPro-7B [10] | DeepSeek-LLM 7B | 1567.1 | 79.2 | 72.1 | 41.0 | 50.0 |
| BLIP3-o 4B [8] | Qwen2.5-VL 3B | 1527.7 | 78.6 | 73.8 | 46.6 | 60.1 |
| BILP3-o 8B [8] | Qwen2.5-VL 7B | 1682.6 | 83.5 | 77.5 | 50.6 | 66.6 |
| BIFROST-1 (Ours) | Qwen2.5-VL 7B | 1685.2 | 83.5 | 76.9 | 58.6 | 66.6 |

Table 3: Comparison with state-of-the-art methods on multimodal generation benchmarks.

| Method | Base (M)LLM | Train Steps×B.S. | COCO FID↓ | MJHQ FID↓ | GenEval↑ | DPG-Bench↑ |
|---|---|---|---|---|---|---|
| EMU [68] | LLaMA 13B | - | 11.66 | - | - | - |
| DreamLLM [18] | Vicuna 7B | - | 8.46 | - | - | - |
| Chameleon [62] | From Scratch 7B | - | 26.74 | - | 0.39 | - |
| Show-o-512 [73] | Phi-1.5 1.3B | - | 9.24 | 15.18 | 0.68 | - |
| VILA-U [72] | LLaMA-2 7B | - | - | 7.69 | - | - |
| EMU3 [68] | From Scratch 7B | - | 12.80 | - | 0.66 | 80.60 |
| MetaMorph [63] | LLaMA-3 8B | - | 11.8 | - | - | - |
| MetaQuery-L [51] | Qwen2.5-VL 3B | - | 8.87 | 6.35 | 0.78 | 81.10 |
| MetaQuery-XL [51] | Qwen2.5-VL 7B | 200M | 8.69 | 6.02 | 0.80 | 82.05 |
| TokenFlow-XL [24] | Qwen-2.5 14B | - | - | - | 0.63 | 73.38 |
| Transfusion [80] | From Scratch 7B | - | 8.70 | - | 0.63 | - |
| LMFusion [59] | LLaVA-Next 8B | - | 8.20 | - | - | - |
| Janus [71] | DeepSeek-LLM 1.5B | 100M | 8.53 | 10.10 | 0.61 | - |
| JanusFlow [43] | DeepSeek-LLM 1.5B | 211M | - | 9.51 | 0.63 | 80.09 |
| JanusPro-1B [10] | DeepSeek-LLM 1.5B | 200M | - | 14.33 | 0.73 | 82.63 |
| JanusPro-7B [10] | DeepSeek-LLM 7B | 194M | - | 13.48 | 0.80 | 84.19 |
| BLIP3-o 4B [8] | Qwen2.5-VL 3B | - | - | - | 0.81 | 79.36 |
| BLIP3-o 8B [8] | Qwen2.5-VL 7B | - | - | - | 0.84 | 81.60 |
| BIFROST-1 (Ours) | Qwen2.5-VL 3B | 9M | 23.02 | 15.24 | 0.61 | 76.41 |
| BIFROST-1 (Ours) | Qwen2.5-VL 7B | 62M | 34.35 | 16.21 | 0.81 | 77.67 |

capabilities of the MLLM and easily scale to backbone MLLMs with larger sizes and stronger performance. Table 2 compares the Qwen2.5-VL 7B [5] model with other unified models on image understanding benchmarks, including MME-P [21], MMB [39], SEED [34], MMMU [76], and MM-Vet [75]. Baseline models such as JanusPro [10] and MetaMorph [63] fully fine-tune the LLM backbone and therefore tend to lose the original planning and reasoning abilities that these LLMs acquired through large-scale text pretraining.

**BIFROST-1 achieves competitive visual generation quality.** In Fig. 7, we compare BIFROST-1 with the recent SoTA methods (*e.g.*, JanusPro and MetaQuery) on MJHQ30k prompts. Table 3 compares BIFROST-1 with other unified models on image generation benchmarks, including COCO and MJHQ for visual quality and GenEval and DPG-Bench for prompt following ability. Trained only with 25M image-text pairs for less than two epochs, BIFROST-1 matches the performance with models trained with much higher compute, including Janus on GenEval benchmark, and outperforms TokenFlow-XL on DPG-Bench. In addition, we would like to highlight that FID scores heavily depend on the choice of diffusion backbones. As also observed in MetaQuery[51], diffusion models fine-tuned on aesthetic datasets (*e.g.*, FLUX.1-dev) typically achieve worse FID scores compared to models that have not undergone extensive aesthetic fine-tuning (*e.g.*, SD1.5, SANA).

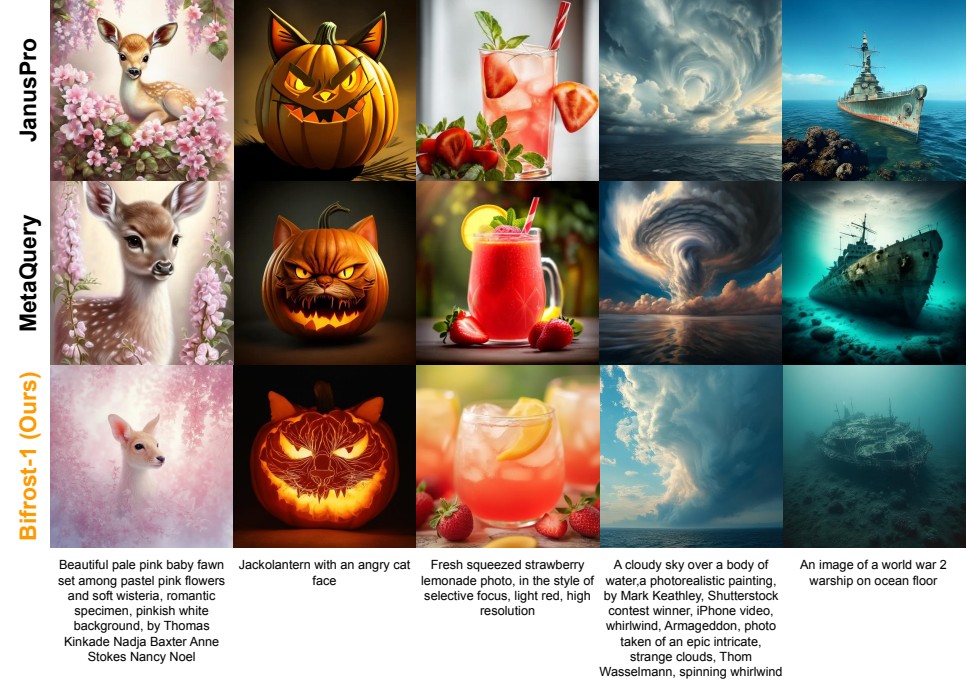

Figure 7: Qualitative comparision with SoTA methods for image generation on MJHQ30k [35].

Table 4: Inference clock time and image generation quality with different VLM decoding steps.

| # Decoding Steps | 64 (default) | 32 | 16 | 8 | 4 | 2 | 1 |
|---|---|---|---|---|---|---|---|
| Clock Time (seconds) | 5.21s | 2.63s | 1.37s | 0.66s | 0.35s | 0.19s | 0.09s |
| FID ↓ | 18.64 | **18.49** | 18.79 | 18.89 | 19.97 | 24.90 | 60.35 |
| sFID ↓ | 43.64 | **43.58** | 44.52 | 45.89 | 50.86 | 67.23 | 217.50 |
| IS ↑ | 156.01 | **158.93** | 156.96 | 158.87 | 150.56 | 124.78 | 83.22 |

## 5.4 MLLM Inference Time with Varying Decoding Steps

To further assess the effect of decoding steps, we conducted experiments with varying numbers of decoding passes on ImageNet. The results are summarized in Tab. 4. We show the inference clock time in seconds, as well as image quality metrics including FID, sFID, and IS. As shown, our method remains robust as long as the number of decoding steps is greater than 8. And we can observe that there is a clear trade-off between inference speed and image quality when using fewer decoding steps. Importantly, we also highlight that the MLLM decoding time is significantly smaller than the runtime of the diffusion-based image generation model. For instance, FLUX.1-dev with 12B parameters together with latent ControlNets takes 14.79 seconds to generate a single image with the default 28 denoising steps. Therefore, the MLLM decoding time is not a major bottleneck. From a practical standpoint, users can flexibly select the number of decoding steps based on whether they favor faster inference or higher image quality for their particular application.

## 6 Conclusion

We present BIFROST-1, a unified and efficient framework for both multimodal understanding and generation. BIFROST-1 bridges pretrained MLLMs with diffusion models by introducing patch-level CLIP image embeddings as latent variables, which are naturally aligned with the CLIP visual encoder of the MLLM. This design enables high-fidelity, controllable image generation while maintaining strong training efficiency. A key advantage of BIFROST-1 is that it fully preserves the original parameters of the MLLM backbone, ensuring that its multimodal planning and reasoning capabilities remain intact. Although our current model is trained with significantly fewer computational resources than SoTA methods and does not yet surpass them in performance, future work focusing on scaling to larger datasets and more powerful MLLM backbones can further realize its potential.

## Acknowledgments

This work was supported by DARPA ECOLE Program No. HR00112390060, NSF-AI Engage Institute DRL-2112635, ARO Award W911NF2110220, ONR Grant N00014-23-1-2356, and a Bloomberg Data Science PhD Fellowship. The views contained in this article are those of the authors and not of the funding agency.

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

# Appendix

## A  Formal Description of the BIFROST-1 MLLM Architecture

As illustrated in Fig. 2, we initialize the visual generation branch from the pretrained MLLM by creating a trainable copy of the MLP and attention QKV projection layers [65]. The only component randomly initialized is the vision head, which is a simple linear projection layer. By reusing the majority of parameters from the pretrained MLLM, we avoid the costly process of realigning image embeddings.

Specifically, we use $\boxed{\text{blue}}$ color to denote the original *frozen* MLLM-specific modules, which handle language modeling (**LM**) and image understanding (**Img-U**) tasks. These modules remain unchanged during training. In addition, we use $\boxed{\text{yellow}}$ color to represent the newly introduced *trainable* modules for the image generation task (**Img-G**), which are initialized as trainable copies of the corresponding blue modules.

During the image generation training process, we first obtain input hidden states for each task. Text inputs $x^{\text{Text}}$ are projected through a linear embedding layer to produce $h_{\text{in}}^{\text{Text}}$. Images used for image understanding ($x^{\text{Img-U}}$) and image generation ($x^{\text{Img-G}}$) are both passed through the frozen MLLM-embedded visual encoder $\mathcal{E}^{\text{Und}}$, producing hidden states $h_{\text{in}}^{\text{Img-U}}$ and $h_{\text{in}}^{\text{Img-G}}$ respectively:

$$h_{\text{in}}^{\text{Text}} = \boxed{\text{Linear}_{\text{Text}}}\left(x^{\text{Text}}\right) \qquad h_{\text{in}}^{\text{Img-U}} = \boxed{\mathcal{E}^{\text{Und}}}\left(x^{\text{Img-U}}\right) \qquad h_{\text{in}}^{\text{Img-G}} = \boxed{\mathcal{E}^{\text{Und}}}\left(x^{\text{Img-G}}\right)$$

During the image generation inference process, $h_{\text{in}}^{\text{Img-G}}$ is initialized from 2D learnable mask tokens.

*By using patch-level CLIP image embeddings that are natively aligned with the MLLM's visual encoder to represent visual signals in vision generation tasks, we eliminate the need for any additional alignment between the visual generation representation and the MLLM.*

For attention processing, we use the frozen MLLM attention layers to compute Q, K, and V matrices for the text and image understanding tokens $h_{\text{in}}^{\text{Text}}$ and $h_{\text{in}}^{\text{Img-U}}$, and use the newly added visual generation branch to process the image generation tokens $h_{\text{in}}^{\text{Img-G}}$:

$$h_{\text{Q}}^{\text{Text}}, h_{\text{K}}^{\text{Text}}, h_{\text{V}}^{\text{Text}} = \boxed{\text{QKV}_{\text{MLLM}}}\left(h_{\text{in}}^{\text{Text}}\right) \qquad h_{\text{Q}}^{\text{Img-U}}, h_{\text{K}}^{\text{Img-U}}, h_{\text{V}}^{\text{Img-U}} = \boxed{\text{QKV}_{\text{MLLM}}}\left(h_{\text{in}}^{\text{Img-U}}\right)$$

$$h_{\text{Q}}^{\text{Img-G}}, h_{\text{K}}^{\text{Img-G}}, h_{\text{V}}^{\text{Img-G}} = \boxed{\text{QKV}_{\text{Img-G}}}\left(h_{\text{in}}^{\text{Img-G}}\right)$$

For language modeling and image understanding tasks, we replicate the standard MLLM attention structure by attending over their respective modalities only. For image generation task, we enable cross-module attention, allowing image generation queries to attend jointly over all token types. Specifically:

$$h_{\text{O}}^{\text{Text}} = \boxed{\text{O}_{\text{MLLM}}}\left(\text{Attn}(h_{\text{Q}}^{\text{Text}}, [h_{\text{K}}^{\text{Text}} \circ h_{\text{K}}^{\text{Img-U}}], [h_{\text{V}}^{\text{Text}} \circ h_{\text{V}}^{\text{Img-U}}])\right)$$

$$h_{\text{O}}^{\text{Img-U}} = \boxed{\text{O}_{\text{MLLM}}}\left(\text{Attn}(h_{\text{Q}}^{\text{Img-U}}, [h_{\text{K}}^{\text{Img-U}} \circ h_{\text{K}}^{\text{Text}}], [h_{\text{V}}^{\text{Img-U}} \circ h_{\text{V}}^{\text{Text}}])\right)$$

$$h_{\text{O}}^{\text{Img-G}} = \boxed{\text{O}_{\text{Img-G}}}\left(\text{Attn}(h_{\text{Q}}^{\text{Img-G}}, [h_{\text{K}}^{\text{Img-G}} \circ h_{\text{K}}^{\text{Img-U}} \circ h_{\text{K}}^{\text{Text}}], [h_{\text{V}}^{\text{Img-G}} \circ h_{\text{V}}^{\text{Img-U}} \circ h_{\text{V}}^{\text{Text}}])\right)$$

where $\circ$ denotes concatenation, and $O(.)$ denotes linear output projection layer. We apply a causal mask to text and image understanding tokens, and a bidirectional mask to image generation tokens.

For the MLP layers, we follow the same branching: text and image-understanding tokens $h_{\text{in}}^{\text{Text}}$ and $h_{\text{in}}^{\text{Img-U}}$ are passed through the frozen MLLM MLP, while image generation tokens $h_{\text{in}}^{\text{Img-G}}$ use the trainable MLP from the generation branch:

$$h_{\text{MLP}}^{\text{Text}} = \boxed{\text{MLP}_{\text{MLLM}}}\left(h_{\text{O}}^{\text{Text}}\right) \qquad h_{\text{MLP}}^{\text{Img-U}} = \boxed{\text{MLP}_{\text{MLLM}}}\left(h_{\text{O}}^{\text{Img-U}}\right) \qquad h_{\text{MLP}}^{\text{Img-G}} = \boxed{\text{MLP}_{\text{Img-G}}}\left(h_{\text{O}}^{\text{Img-G}}\right)$$

Finally, task-specific heads convert the hidden states to output predictions. For language modeling and image understanding, we apply a linear projection to $h_{\text{MLP}}^{\text{Text}}$, and for image generation, we project $h_{\text{MLP}}^{\text{Img-G}}$ using the vision generation head:

$$h_{\text{out}}^{\text{Text}} = \boxed{\text{TextHead}}\left(h_{\text{MLP}}^{\text{Text}}\right) \qquad\qquad h_{\text{out}}^{\text{Img-G}} = \boxed{\text{VisionHead}}\left(h_{\text{MLP}}^{\text{Img-G}}\right)$$

## B Broader Impacts

BIFROST-1 is motivated by the fact that training a unified multimodal generation and understanding model that can perform native generation with high visual quality usually requires huge computational cost. By bridging pretrained MLLM with pretrained diffusion models, training BIFROST-1 can be significantly faster. Therefore, we believe that our work can be a strong contribution to efficient unified model training. While our framework can benefit numerous applications in image generation, similar to other image generation frameworks, it can also be used for potentially harmful purposes (e.g., creating false information or misleading images). Therefore, it should be used with caution in real-world applications.

## C Safeguards

BIFROST-1 is built upon pretrained MLLM (*i.e.*, Qwen2.5-VL) and diffusion models (*i.e.*, FLUX.1-dev) with strong safeguards, and trained on publically available image datasets (*i.e.*, MSCOCO and SA1B) that removes unsafe concepts. Therefore, our model avoids the high risk for misuse.

## D Limitations

Note that BIFROST-1 is designed as a bridging method that connects existing MLLMs with diffusion-based image generation models. As such, its performance, output quality, and potential visual artifacts are inherently influenced by the capabilities and limitations of the underlying backbone models it relies on. For instance, if the diffusion model used as the visual backbone struggles with generating complex, rare, or previously unseen scenes and objects, then BIFROST-1, which builds upon this foundation, may also exhibit suboptimal image generation results. This dependency highlights the importance of selecting strong and well-generalized base models when applying BIFROST-1 to real-world or open-domain generation tasks.

## E License

We use standard licenses from the community and provide the following links to the licenses for the datasets, codes, and models that we used in this paper. For further information, please refer to the specific link.

PyTorch [4]: BSD-style

HuggingFace Transformers [70]: Apache License 2.0

HuggingFace Diffusers [66]: Apache License 2.0

FLUX.1-dev [32]: Non-Commercial License

Qwen2.5-VL [5]: Non-Commercial License

MSCOCO dataset [38]: CC BY 4.0

CC12M dataset [7]: Permissive Custom License

SA1B dataset [30]: SA-1B Dataset Research License

MJHQ30k dataset [35]: Playground v2 Community License

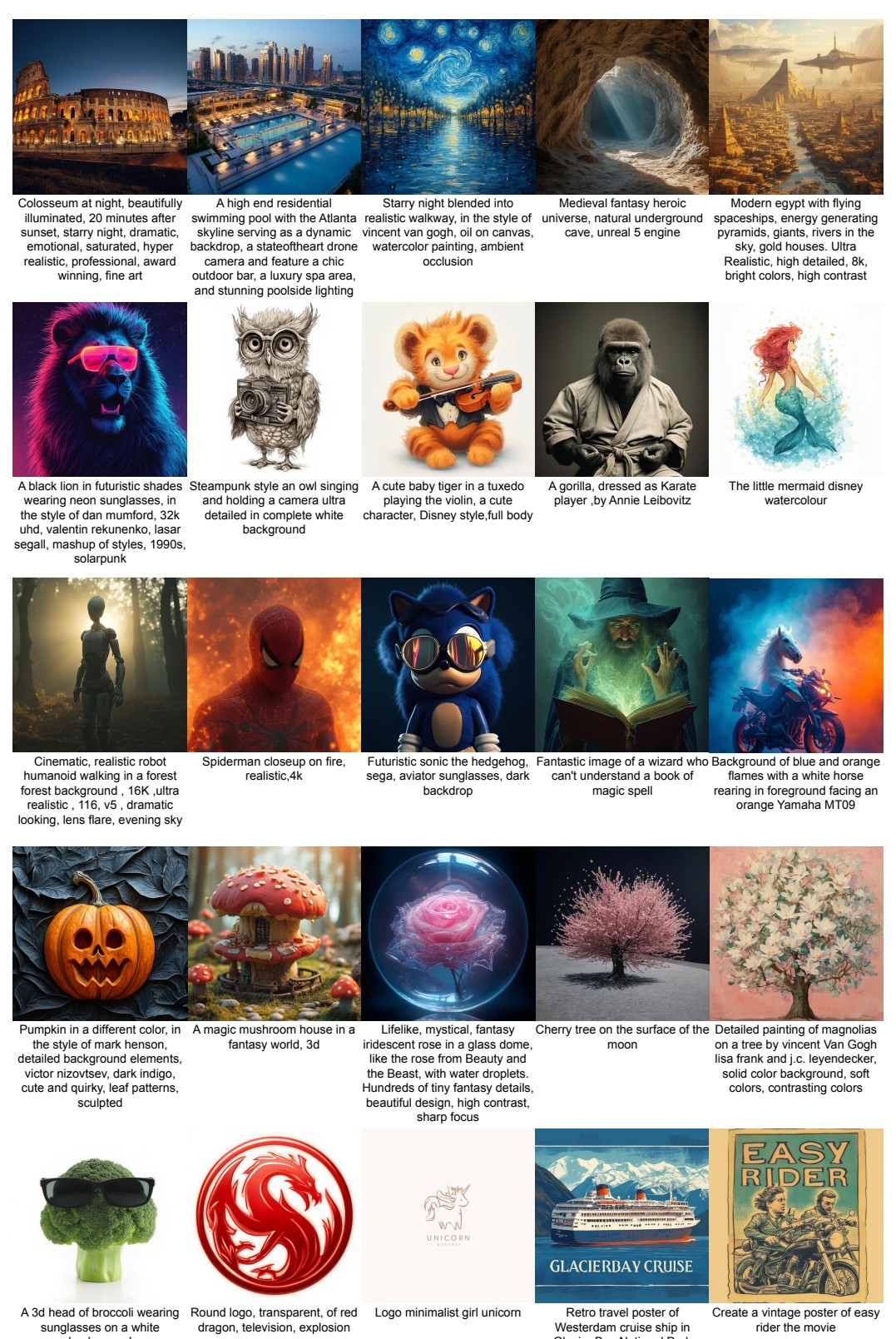

Figure 8: Visualization examples from MJHQ30k dataset.

