# OpenReview forum: "Bifrost-1: Bridging Multimodal LLMs and Diffusion Models with Patch-level CLIP Latents"
_NeurIPS.cc/2025/Conference — NeurIPS 2025 poster_

### Official Review · Reviewer_XePf · 2025-07-01

**Clarity:** 3
**Significance:** 2
**Originality:** 2
**Rating:** 4
**Confidence:** 2

**Summary:**

This paper proposes a novel framework for a unified generation-understanding task model. The framework is built upon a pretrained MLLM and a diffusion-based image generator, with the addition of two modules: a Vision Generation branch and a patch-level latents ControlNet. The authors claim that this framework preserves the reasoning capabilities of the MLLM while also maintaining training efficiency.

**Questions:**

* As indicated in lines 203-210, why were different training steps used in the experiments in sections 5.1-5.3?

* Are there any additional scale-up results, such as using Qwen2.5-VL 7B as the base model?

**Ethical Concerns:**

["NO or VERY MINOR ethics concerns only"]

**Final Justification:**

I would like to thank the authors for the detailed response. Some of my concerns have been addressed, and I will keep my score unchanged.

**Limitations:**

yes

**Paper Formatting Concerns:**

None.

**Quality:**

2

**Strengths And Weaknesses:**

## Strengths

* The experiments show that the proposed architecture injects image generation capabilities with a relatively small training cost while maintaining the MLLM inference ability.

* The paper is well-written and easy to understand.

## Weakness

* The main difference between the proposed method and the one in Fig. 1(b) seems to be the change from 1D tokens to 2D CLIP tokens for the input diffusion model's latent. This appears somewhat trivial. Could you elaborate on the differences between the method proposed in this paper and the one in Fig. 1(b)?

* In Table 2, the proposed method and MetaQuery-L seem to have comparable performance, with no significant improvement in performance.

---

> ### Author Rebuttal · Authors · 2025-07-31
>
> We would like to sincerely thank the Reviewer for the feedback.
>
> > ### W1: The main difference between the proposed method and the one in Fig. 1(b)
>
> We sincerely thank the reviewer for raising this important comparison with the related works in Fig.1 (b). Below are several key differences between our method and the methods under Fig.1 (b):
>
> - **Image Prior Injection Strategy:** The key difference between our method and those in Fig. 1(b) is that we directly inject 2D image tokens from the MLLM into DiT latents via a ControlNet-based architecture, enabling stronger and more explicit spatial conditioning. In contrast, Fig. 1(b) methods use image/text tokens (e.g., MetaMorph, MetaQuery, DreamLLM) or bounding box layouts (e.g., Layout-GPT, VPGen), typically injecting information via cross-attention, which offers weaker spatial guidance and lower training efficiency.
>
> To support our claim that Latent ControlNet is more efficient than cross-attention, we conducted an additional experiment comparing our default ControlNet-based injection with a cross-attention-based variant using the same 64 MLLM-generated image tokens. The cross-attention setup unfreezes all modules in the conditioning branch, resulting in 21% trainable parameters in FLUX.1-dev, compared to only 13% in our design. Both variants were trained on ImageNet for 7 epochs under the same compute budget. As shown below, the ControlNet-based approach achieves better image generation quality (FID, sFID, IS), demonstrating its superior efficiency and effectiveness.
>
> | Condition Injection Strategy | FID (↓) | sFID (↓) | IS (↑)  |
> | :---- | :---- | :---- | :---- |
> | Latent ControlNet (Ours) | **19.21** | **47.08** | **137.25** |
> | Cross Attention  | 76.32   | 208.11 | 26.35 |
>
> - **Training Objective:** Some methods in Fig. 1(b), such as DreamLLM and MetaMorph, generate visual tokens autoregressively (AR) and visualize them using a diffusion head. In contrast, we adopt a discrete diffusion / MAR [1] decoding strategy, where masked tokens are iteratively decoded into clean tokens. During inference, we randomly sample a generation order over image patches and follow a MaskGIT-style [2] cosine schedule to progressively unmask tokens. This schedule, as visualized in Figure 2 of the MaskGIT paper, allocates fewer unmasked tokens in the early stages (when uncertainty is high), and progressively more in later steps as more tokens become available. As shown in Table 1 of the MAR paper, MAR is significantly more efficient and parallelizable than standard AR, supporting our design decision to prioritize training efficiency.
>
> - **MLLM-aligned encoder (Ours) vs. No pre-aligned visual encoder:** Our method uses the MLLM’s native CLIP visual encoder for visual generation, while most methods in Fig. 1(b) use non–pre-aligned visual encoders such as SigLIP or VAE. Our approach offers three key advantages:
>
>   - (1) Since we initialize the visual generation branch with the same parameters as the original MLLM, its features are natively aligned with the MLLM’s CLIP visual encoder, which significantly accelerates training.
>
>   - (2) Unlike methods that adopt new visual encoders (e.g., SigLIP or VAE), our approach does not require additional projection layers to connect the MLLM with the visual encoder, simplifying the architecture and improving efficiency.
>
>   - (3) The MLLM’s native CLIP encoder encodes a 256×256 image into just 64 patch tokens. In contrast, non–pre-aligned encoders like SigLIP (e.g., ViT-SO400M-14-SigLIP-384) upscale the image to 384×384 and produce 729 patch tokens, requiring further downsampling or compression. This adds complexity and may lead to information loss.
>
> To validate our claims, we compare our default strategy (using the MLLM’s native CLIP visual encoder) with the SigLIP encoder under the same compute budget (7 epochs for the MLLM vision branch and 1 epoch for ControlNet) on ImageNet at 256×256 resolution. Our method consistently outperforms the SigLIP-based variant in image quality. For completeness, we also include results with the VAE encoder from FLUX.1-dev, as shown in Table 1 of our main paper, which further highlights our approach’s efficiency and performance advantage.
>
> | Visual Encoder | FID (↓) | sFID (↓) | IS (↑) |
> | :---- | :---- | :---- | :---- |
> | MLLM’s Native CLIP Visual Encoder (Ours)  | **19.21** | **47.08** | **137.25** |
> | SigLIP (e.g., MetaMorph) | 274.16 | 304.94    | 2.69 |
> | VAE Encoder (from FLUX.1-Dev) | 284.51 | 361.45 | 1.11 |
>
> - **Parameter-efficient training (Ours) vs. full-finetuning:** We insert lightweight ControlNet blocks (13% of FLUX.1-dev) into a frozen DiT, keeping all pretrained weights untouched. In contrast, methods in Fig. 1(b) often fully fine-tune DiT to adapt to new visual tokens, which becomes especially challenging for video generation tasks requiring high fidelity and temporal consistency.
>
> - **MLLM Freezing Strategy:** We keep the MLLM fully frozen to preserve its reasoning and alignment abilities. In contrast, methods like MetaMorph and JanusPro [3] fine-tune the MLLM to enable token generation, risking degraded performance without massive, high-quality data. As shown in Table 3 of the JanusPro paper, their multimodal understanding performance is much lower than that of MetaQuery and BLIP3o, both of which keep the backbone frozen. This supports our design choice to freeze the MLLM.
>
> We hope this helps clarify our technical distinctions from Fig. 1(b). We’re happy to provide further clarification if needed. Thank you again!
>
> [1] Autoregressive image generation without vector quantization, NeurIPS, 2024
>
> [2] Maskgit: Masked generative image transformer, CVPR, 2022.
>
> [3] Janus-pro: Unified multimodal understanding and generation with data and model scaling, arXiv, 2025.
>
> ---
>
> > ### W2: The proposed method and MetaQuery-L seem to have comparable performance on multimodal understanding tasks
>
> Thank you for this insightful observation! Indeed, the concurrent work MetaQuery and our method achieve the same scores on multimodal understanding tasks, as both use Qwen2.5-VL as the base model and keep it frozen during training. Specifically, MetaQuery uses query tokens to extract semantic information from the frozen multimodal LLM, while our method creates a trainable copy as the vision branch, sharing the same visual encoder. As a result, both approaches are expected to perform similarly on understanding tasks. We will clarify this point in the camera-ready version. Thank you again for highlighting this important detail.
>
> ---
>
> > ### Q1: Clarification on the experiments in sections 5.1-5.3
>
> Thank you very much for raising this question. Sections 5.1 and 5.2 are conducted on the ImageNet dataset at 256×256 resolution, which contains approximately 1.2M images across 1,000 classes. Specifically, Section 5.1 focuses on text-to-image generation, while Section 5.2 presents image reconstruction experiments. All experiments in these sections are trained on a single GB200 GPU.
>
> In Section 5.3, we scale up to open-world image-text pairs, using a total of 5M samples. Accordingly, we increase the training resources to 8 H100 GPUs instead of a single GB200.
>
> Additionally, we observe that our latent ControlNet typically converges much faster than the vision branch in multimodal LLMs due to its lightweight design. As a result, we allocate a smaller training budget to ControlNet compared to the MLLM vision branch.
>
> ---
>
> > ### Q2: Additional scale-up results
>
> Thank you very much for the valuable question regarding training cost. In addition to the comparison of different architectures under the same compute budget in Table 1 of our main paper, we would like to highlight that training cost comparisons with other existing baselines are also provided in Appendix Table 1 (available in the PDF of the supplementary materials, as mentioned in L284-285 of our main paper). In Appendix Table 1, training cost is measured by the total amount of data used, calculated as the product of the number of training steps and the batch size per step. As shown, our method achieves performance comparable to baselines such as Janus and EMU, while requiring significantly less training data (9M for Ours vs. 100M for Janus).
>
> Beyond the results presented in the appendix, we obtained additional GPU resources after the submission deadline and conducted larger-scale experiments using the same training dataset as BLIP3-o [4], which is a recent SoTA method that connects MLLMs with diffusion models for image generation (released on May 14, 2025). Specifically, we use 27 million image-text pairs from the BLIP3o-Long-Caption dataset and train our model for 1.7 epochs, followed by fine-tuning on 60K high-quality images from the BLIP3o-60K dataset. In contrast, BLIP3-o is trained on the same 27M dataset for over 10 epochs. A comparison is summarized in the table below:
>
> | Models | MLLM Backbone | Total Pretraining Images (Training Steps×Batch Size) | GenEval | DPGBench |
> | :---- | :---- | :---- | :---- | :---- |
> | BLIP3-o | Qwen2.5-VL 3B | \~270M | 81% | 79.36% |
> | BLIP3-o | Qwen2.5-VL 7B | \~270M | 84% | 81.60% |
> | Bifrost-1 (Ours) | Qwen2.5-VL 3B | 9M | 61% | 76.41% |
> | Bifrost-1 (Ours) | Qwen2.5-VL 7B | 46M | 81% | 77.67% |
>
> As shown, the GenEval score improves from 61% to 81%, and DPGBench increases from 76.41% to 77.67%, demonstrating that better annotations and scaled-up training lead to meaningful gains. Notably, even with this larger-scale setup, our training cost remains much lower than prior works, while still outperforming existing approaches under the same compute budget.
>
> [4] Blip3-o: A family of fully open unified multimodal models—architecture, training and dataset, arXiv, 2025.
>
> ---
>
> Dear Reviewer XePf,
>
> We sincerely thank you for all the comments and valuable feedback. We believe that we have addressed all the questions in depth. If you have any additional questions, please feel free to let us know, and we would be more than happy to address them.

---

> > ### Comment · Reviewer_XePf · 2025-08-05
> >
> > Thank you for the detailed response. Some of my concerns have been addressed, and I will keep my score unchanged.

---

> > ### Author Response · Authors · 2025-08-06
> >
> > Dear Reviewer XePf,
> >
> > We would like to once again express our sincere thanks for acknowledging that “our proposed architecture injects image generation capabilities with a relatively small training cost while maintaining the MLLM inference ability.”
> >
> > We’re glad that some of your concerns have been addressed, and we would be happy to clarify any remaining questions to help you reconsider your evaluation. If there is anything else that remains unclear, we are more than willing to provide further explanation.
> >
> > Finally, we truly appreciate your time and thoughtful engagement throughout this rebuttal process. Thanks!

---

> ### Author Response · Authors · 2025-08-08
> **Sincere thank for the Reviewer's feedback**
>
> Dear Reviewer XePf,
>
> As today marks the end of the author-reviewer discussion period, we would like to express our heartfelt thanks for your valuable feedback and thoughtful suggestions throughout the rebuttal process.
>
> We sincerely appreciate your recognition of our work, particularly our proposed architecture design that injects image generation capabilities with a relatively small training cost while maintaining the MLLM inference ability.
>
> It’s also glad to know that our previous response could be helpful to address your concerns. If you have any remaining questions, please don’t hesitate to let us know, and we would be more than happy to address them by the end of today.
>
> Thank you again for your time and effort!

---

### Official Review · Reviewer_k2H1 · 2025-07-02

**Clarity:** 3
**Significance:** 3
**Originality:** 3
**Rating:** 4
**Confidence:** 3

**Summary:**

This paper proposes a framework that bridges pretrained MLLM and diffusion models, enabling high-fidelity image generation with strong training efficiency. By preserving the original MLLM parameters, it maintains multimodal reasoning capabilities while achieving competitive performance.

**Questions:**

What happens to generation fidelity and diversity as the number of CLIP tokens increases? In the case of VAE representations, increasing the number of tokens often improves reconstruction but can harm the diversity of generated samples. I wonder if similar effects occur here.

**Ethical Concerns:**

["NO or VERY MINOR ethics concerns only"]

**Final Justification:**

I appreciate authors for their rebuttal. The author addresses most of my concerns so I would like to keep my positive stance on this paper.
But for W1, I understand author's claim, but I hope there could be better if support experiments or evidence are provided.

**Limitations:**

Yes

**Quality:**

3

**Strengths And Weaknesses:**

# Strengths
- The overall performance is solid, and the experimental setup is well-structured. The experiments convincingly support the effectiveness of the proposed method.
- Extensive ablation studies and benchmark evaluations clearly demonstrate the contributions of the proposed modules.
- The design choices are well justified through ablation studies, including comparisons between CLIP latents and other alternatives like VAE or learnable 2D query tokens.

# Weaknesses
- Bifrost-1 incorporates a latent ControlNet to bridge patch-level CLIP latents into the diffusion model, but the motivation for introducing a new latent ControlNet structure is not entirely clear. For instance, could an alternative have been to let the MLLM generate a modality (e.g., depth, sketch) that could be consumed by an existing pretrained ControlNet? While this might compromise reconstruction quality, how would it affect generation fidelity?
- Although the paper emphasizes the training efficiency of the proposed method, it would be better to include a quantitative comparison of training costs with existing baselines.
- The explanation of some internal components (e.g. vision generation branch) is lacking. A more detailed architectural description would help clarify how each part contributes to the overall system.

---

> ### Author Rebuttal · Authors · 2025-07-31
>
> We would like to sincerely thank the Reviewer for all the valuable feedback. In the rebuttal below, we use “W1/W2/W3” to represent the answer to the bullet points in weaknesses, and “Q1” to represent the answer to the questions.
>
>
>
> > ### W1: Could an alternative have been to let the MLLM generate a modality (e.g., depth, sketch) that could be consumed by an existing pretrained ControlNet? While this might compromise reconstruction quality, how would it affect generation fidelity
>
> Thank you very much for bringing up this interesting question! Indeed, one could allow the MLLM to generate an intermediate modality such as depth or sketch, and then reuse a pretrained ControlNet together with an image generation model to render the final image. However, this approach may have several limitations:
>
> - First, the information encoded in a depth or sketch image may be less comprehensive than what is captured by MLLM-generated latents (e.g., the 2048-dimensional CLIP latents from Qwen2.5-VL 3B). Such MLLM-aligned CLIP visual encoder makes it easier to capture richer patch-level image semantics compared to the structural cues available in depth or sketch images. For example, the latents can encode more detailed semantics, including object layouts, colors, textures, and shapes—going beyond simple depth or edge information.
>
> - Second, using a pretrained depth/sketch ControlNet can be more error-prone than training a latent-based ControlNet tailored to the MLLM outputs. This is because the depth or sketch images generated by the MLLM might be imperfect. As a result, strictly following such noisy guidance using a pretrained ControlNet can lead to suboptimal image quality. In contrast, training a ControlNet directly on MLLM-generated latents offers the flexibility to learn how to interpret and leverage those more-or-less noisy latents more effectively than relying on fixed modality-specific guidance. In fact, this is precisely one of the main reasons why we connect the MLLM with DiT through ControlNets. We aim to reduce the burden on the MLLM to generate highly accurate images, as high-level semantics or “blurry” latents are already sufficient to guide the image generation model.
>
> ---
>
> > ### W2: Quantitative comparison of training costs with existing baselines
>
> Thank you very much for the valuable question regarding training cost. In addition to the comparison of different architectures under the same compute budget in Table 1 of our main paper, we would like to highlight that training cost comparisons with other existing baselines are also provided in Appendix Table 1 (available in the PDF of the supplementary materials, as mentioned in L284-285 of our main paper). In Appendix Table 1, training cost is measured by the total amount of data used, calculated as the product of the number of training steps and the batch size per step. As shown, our method achieves performance comparable to baselines such as Janus and EMU, while requiring significantly less training data (9M for Ours vs. 100M for Janus).
>
> Beyond the results presented in the appendix, we gained more GPU support after the paper submission deadline, and here we include additional larger-scale experiments trained on the same training dataset from BLIP3-o [1], one of the latest works that connects MLLM with diffusion models for image generation released on May 14, 2025. Specifically, we use 27 million image-text pairs from BLIP3o-Long-Caption dataset, and train the model for 1.7 epochs, followed by fine-tuning on 60k high-quality images in the BLIP3o-60K dataset. In contrast, BLIP3-o is trained on the same 27M dataset for over 10 epochs. We summarize the comparison in the following table:
>
> | Models | MLLM Backbone | Total Pretraining Images (Training Steps×Batch Size) | GenEval | DPGBench |
> | :---- | :---- | :---- | :---- | :---- |
> | BLIP3-o | Qwen2.5-VL 3B | \~270M | 81% | 79.36% |
> | BLIP3-o | Qwen2.5-VL 7B | \~270M | 84% | 81.60% |
> | Bifrost-1 (Ours) | Qwen2.5-VL 3B | 9M | 61% | 76.41% |
> | Bifrost-1 (Ours) | Qwen2.5-VL 7B | 46M | 81% | 77.67% |
>
> As shown, the GenEval score improves significantly from 61% to 81%, and the DPGBench score increases from 76.41% to 77.67%. This demonstrates that higher-quality training data with better annotations, as well as scaled-up training, can lead to meaningful performance gains. Lastly, we would like to emphasize that even with this larger-scale training, our training cost remains much lower than that of prior works, and our method still outperforms previous approaches under the same training budget.
>
> [1] Chen, Jiuhai, et al. "Blip3-o: A family of fully open unified multimodal models—architecture, training and dataset." arXiv preprint arXiv:2505.09568 (2025).
>
> ---
>
> > ### W3: A more detailed architectural description
>
> Thanks for your question. Here we explain our architecture in more detail.
>
> - **Adding a vision generation branch to the MLLM:** Starting from a pretrained MLLM (e.g., Qwen2.5-VL 3B), we create a copy of its parameters (e.g., attention layers, FFN layers) to serve as the visual generation branch. This branch is initialized with the same pretrained weights as the original model. The original MLLM parameters are kept fully frozen, and only the new vision generation branch is trainable.
>
> - **Encoding images using the MLLM’s native visual encoder:** Unlike prior works that use a VAE or CLIP/SigLIP-based encoder to process images, we utilize the MLLM’s native visual encoder during training to encode the images. As shown in Table 1 of our main paper, using the MLLM’s native visual encoder, rather than a VAE as the image encoder, results in significantly better image generation quality under the same computation budget.
>
> - **Masked autoregression (MAR) for visual token prediction:** During training, we apply MAR to teach the model to predict masked visual tokens. Specifically, the training image is first encoded using the MLLM’s visual encoder. Then, a random value α is sampled, and α × (number of visual tokens) are randomly masked. The text condition is passed into the frozen MLLM branch to guide visual token prediction. The model is trained to reconstruct the masked tokens, with the training objective being the mean squared error (MSE) loss between the predicted and ground truth visual tokens in the masked regions.
>
> - **Latent ControlNet for image decoding:** The predicted visual tokens are treated as 2D visual cues or image sketches, which are then passed to a latent ControlNet to generate the final image. The overall architecture closely follows the FLUX.1-Dev ControlNet design, but we make it more lightweight by using only 4 MM-DiT (DoubleStream) blocks and 1 Single-DiT (SingleStream) block (i.e., num_double_layers=4, num_single_layers=1). The training objective matches that of FLUX DiT, using a flow-matching noise scheduler to sample timesteps and predict the flow direction from noise to data.
>
> - **Inference process:** During inference, we begin with a fully masked 2D token grid, which is fed into the MLLM’s vision generation branch. The accompanying text condition is provided to the frozen MLLM branch to guide the visual token prediction. Prior to decoding, we randomly sample a generation order over the image patch tokens, which determines the sequence in which masked tokens are decoded into clean ones. Following MaskGIT [2], we adopt a cosine schedule to determine how many tokens are unmasked at each decoding step. This schedule, as visualized in Figure 2 of the MaskGIT paper, allocates fewer unmasked tokens in the early stages (when uncertainty is high), and progressively more in later steps as more tokens become available. Finally, these decoded tokens are then passed to the latent ControlNet as visual cues to guide final image generation.
>
>
> [2] Chang, Huiwen, et al. "Maskgit: Masked generative image transformer." Proceedings of the IEEE/CVF conference on computer vision and pattern recognition. 2022.
>
> ---
>
> > ### Q1: Generation fidelity and diversity as the number of CLIP tokens increases
>
> Thank you again for raising this interesting question! In our current implementation, we do not use additional modules or loss functions to explicitly encourage diversity in the generated objects. Therefore, the observation from VAEs, where diversity decreases as the number of tokens increases, may also apply to our case.
>
> While this paper primarily focuses on how to effectively connect MLLMs with diffusion models, we fully agree that promoting diversity in the generated outputs is an important and orthogonal direction worth exploring. We are aware of existing works that address this issue [3], and it would be interesting to investigate whether such methods can be adapted to our architecture. We leave this for future work.
>
>
> [3] Gu, Jiatao, et al. "Kaleido Diffusion: Improving Conditional Diffusion Models with Autoregressive Latent Modeling." Advances in Neural Information Processing Systems 37 (2024): 5498–5527.
>
>
> ---
>
> Dear Reviewer k2H1,
>
> Once again, we sincerely thank you for all the comments and valuable feedback. We believe that we have addressed all the questions in depth, including (1) explanation of our design choice of generating image latents and the use of latent ControlNet, (2) quantitative comparison of training costs with existing baselines, (3) a more detailed architectural description, and (4) discussion of the influence of generation fidelity and diversity as the number of CLIP tokens increases. If the reviewer has any additional questions, please feel free to let us know, and we would be more than happy to address them.

---

> ### Author Response · Authors · 2025-08-08
> **Sincere thank for the Reviewer's feedback**
>
> Dear Reviewer k2H1,
>
> As today marks the end of the author-reviewer discussion period, we would like to express our heartfelt thanks for your valuable feedback and thoughtful suggestions throughout the rebuttal process.
>
> We sincerely appreciate your recognition of the strengths of our work. We're grateful for your positive assessment of the overall performance and the well-structured experimental setup, as well as the well-justified design choices, especially the comparisons between CLIP latents and alternatives such as VAE or learnable 2D query tokens.
>
> If you have any remaining questions or concerns, please don’t hesitate to let us know, and we would be more than happy to address them by the end of today.
>
> Thank you again for your time and effort!

---

### Official Review · Reviewer_98F3 · 2025-07-03

**Clarity:** 2
**Significance:** 2
**Originality:** 2
**Rating:** 2
**Confidence:** 5

**Summary:**

This paper proposes Bifrost-1, which bridges the MLLMs with diffusion models by patch-level CLIP latents to make the MLLMs generate images. The core innovation is to use 2D patch-level CLIP latents as the communicative medium, instead of 1D tokens. The original MLLM is duplicated into a two-branch network, in which the new branch is trained to predict patch-level CLIP embeddings. The diffusion model is finetuned by ControlNet to decode the generated visual token into an image. Experiments on understanding and generation benchmarks have shown competitive results.

**Questions:**

1. Due to the core innovation of 2D patch-level CLIP latents appearing to overlap significantly with MetaMorph, the authors may clarify what the significant differences are between these two methods. A clear analysis against MetaMorph would help clarify the novelty.

2. Since the MLLMs already have understanding capabilities, the core evaluation should be image generation. Therefore, I suggest moving the generation benchmark results from the appendix into the main paper to emphasize their importance

**Ethical Concerns:**

["NO or VERY MINOR ethics concerns only"]

**Final Justification:**

After multiple rounds of discussions, we still have fundamentally different views regarding the contributions of this paper.

1. I am not convinced by the newly claimed contributions and new experiments provided in the author responses. My reasons are detailed in our previous discussions, which are well-supported by related works. Based on this assessment, I recommend rejection. I kindly ask the Area Chairs to read these discussions before making a decision.

2. Even if I were to tentatively accept the newly claimed contributions **(which I do not)**, they differ significantly from the original central claim of "Patch-level CLIP Latents". The current paper is structured around that original claim. Accepting it would require a **substantial rewrite** of the paper to properly reflect its true contributions. Such a situation is not suitable for a conference submission, as conferences do not provide a mechanism to supervise or ensure this level of rewriting.

Given these points, I do not believe the paper is ready for publication at a top-tier venue such as NeurIPS.

**Limitations:**

yes

**Quality:**

2

**Strengths And Weaknesses:**

Strengths

1. Bifrost-1 proposes to use 2D patch-level CLIP latents as the communicative medium between the MLLM and diffusion models, which can convey more detailed visual information for high-quality generation.

2. The network is trained efficiently and shows reasonable understanding and generation results.

Weaknesses

1. My main concern about this paper is its significant conceptual overlap with MetaMorph.

1a) Bifrost-1 claims 2D patch-level CLIP latents as its core innovation. However, MetaMorph already uses fine-grained CLIP latents (64 tokens to regress SigLIP image embeddings), as shown in the MetaMorph paper Section 2.1: Tokenizing multimodal data. Therefore, I think the core innovation is not convincing enough.

1b) Both the Bifrost-1 and MetaMorph use the continuous GT CLIP embedding to supervise the generated visual tokens, making the learning objective similar.

1c) Figure 1b misrepresents MetaMorph by labeling it as using 1D image tokens. However, since MetaMorph visual tokens regress SigLIP image embeddings, it should be 2D image tokens. Moreover, the network architecture in Figure 1c is very similar to the structure in Figure 1 in MetaMorph, in which the LLM predicts fine-grained CLIP image tokens to condition the diffusion model.

2. The core motivation of this paper is to enable strong image generation capabilities of MLLMs. However, the results on COCO FID, MJHQ FID, GenEval, and DPG-Bench in Table 1 in the Appendix are not as competitive as previous methods, which undermines its central claim of enabling high-quality generation.

---

> ### Author Rebuttal · Authors · 2025-07-31
>
> We would like to sincerely thank the Reviewer for all the valuable feedback. In the rebuttal below, we use “W1/W2” to represent the answer to the bullet points in weaknesses, and “Q1/Q2” to represent the answer to the questions.
>
> > ### W1/Q1: Comparison with MetaMorph
>
> We sincerely thank the reviewer for raising this important comparison with the related work, MetaMorph. We fully agree that both MetaMorph and our method share the high-level goal of developing a unified framework that decodes LLMs/VLMs outputs with diffusion models. However, we would like to kindly emphasize several key differences between the two approaches:
>
> - **Image Prior Injection Strategy:** Our method directly injects 2D image tokens generated by the MLLM into the DiT latents via a ControlNet-based architecture. By adding 2D ControlNet latents on top of the noisy DiT latents, our method can enforce spatial structures more explicitly and effectively. In contrast, MetaMorph uses cross-attention to condition on image tokens. which behaves more like a global reference image. However, such cross-attention conditioning typically provides weaker spatial guidance than ControlNet and results in lower training efficiency.
>
> To support our claim that Latent ControlNet is more efficient than cross-attention, we conducted an additional experiment comparing our default ControlNet-based injection with a cross-attention-based variant using the same 64 MLLM-generated image tokens. The cross-attention setup unfreezes all modules in the conditioning branch, resulting in 21% trainable parameters in FLUX.1-dev, compared to only 13% in our design. Both variants were trained on ImageNet for 7 epochs under the same compute budget. As shown below, the ControlNet-based approach achieves better image generation quality (FID, sFID, IS), demonstrating its superior efficiency and effectiveness.
>
> | Condition Injection Strategy | FID (↓) | sFID (↓) | IS (↑)  |
> | :---- | :---- | :---- | :---- |
> | Latent ControlNet (Ours) | **19.21** | **47.08** | **137.25** |
> | Cross Attention  | 76.32   | 208.11 | 26.35 |
>
> - **Training Objective:** MetaMorph generates visual tokens autoregressively (AR) and then visualizes them using a diffusion head. In contrast, we adopt a discrete-diffusion/MAR [1] decoding strategy to decode from masked tokens into clean visual tokens. Specifically, during inference, we randomly sample a generation order over the image patch tokens, which determines the sequence in which masked tokens are decoded into clean ones. Following MaskGIT [2], we adopt a cosine schedule to determine how many tokens are unmasked at each decoding step. This schedule, as visualized in Figure 2 of the MaskGIT paper, allocates fewer unmasked tokens in the early stages (when uncertainty is high), and progressively more in later steps as more tokens become available. As shown in Table 1 of the MAR paper, MAR is significantly more efficient and parallelizable than standard AR, supporting our design decision to prioritize training efficiency. This training objective difference, together with the cross-attention image token injection strategy, made us to classify MetaMorph into Fig1(b). This being said, we fully agree with the reviewer's suggestion that classifying MetaMorph as a 1D image token may be inaccurate. We will update the label of Fig. 1(b) to 'Bridge w/ 1D/2D Image/Text Tokens' in the camera-ready version. Thank you again for pointing out this inaccuracy.
>
> - **MLLM-aligned encoder (Ours) vs. No pre-aligned visual encoder (MetaMorph):** Our method uses the MLLM’s native CLIP visual encoder for visual generation, while MetaMorph uses SigLIP (i.e., ViT-SO400M-14-SigLIP-384) to encode images. Our approach offers three key advantages:
>
>   - (1) Since we initialize the visual generation branch with the same parameters as the original MLLM, the features in the visual generation branch are natively aligned with the MLLM’s CLIP visual encoder.
>
>   - (2) Compared to using new visual encoders such as SigLIP or VAE, our method does not require learning additional projection layers to connect the MLLM with the visual encoder, simplifying the architecture and improving efficiency.
>
>   - (3) The MLLM’s native CLIP encoder encodes a 256×256 image into just 64 patch tokens. In contrast, ViT-SO400M-14-SigLIP-384 first upscales the image to 384×384 and encodes it into 729 patch tokens, requiring additional downsampling or compression to reduce the number of image tokens. MetaMorph uses vanilla bilinear interpolation to compress the features down to 64 tokens, which may lead to information loss. Alternatively, one would need to use learnable downsampling blocks to better preserve image information, adding architectural complexity.
>
> To support these claims, we provide a new quantitative comparison between our default strategy (MLLM’s native CLIP visual encoder) and the SigLIP encoder. We train both variants with the same compute: 7 epochs for the MLLM vision branch and 1 epoch for ControlNet, on the ImageNet dataset at 256×256 resolution. As shown, our default strategy achieves much better image generation quality compared to the SigLIP-based approach. Additionally, Table 1 in our main paper compares our default strategy with the VAE encoder used in FLUX.1-dev. We also show the numbers for VAE encoder here for completeness.
>
>
> | Visual Encoder | FID (↓) | sFID (↓) | IS (↑) |
> | :---- | :---- | :---- | :---- |
> | MLLM’s Native CLIP Visual Encoder (Ours)  | **19.21** | **47.08** | **137.25** |
> | SigLIP (following MetaMorph)  | 274.16 | 304.94    | 2.69 |
> | VAE Encoder (from FLUX.1-Dev) | 284.51 | 361.45 | 1.11 |
>
> - **Parameter-efficient training (Ours) vs. full-finetuning (MetaMorph):** We insert lightweight ControlNet blocks (13% of FLUX.1-dev) into a frozen DiT, keeping all pretrained weights untouched. In contrast, MetaMorph needs to fine-tune DiT to adapt to new visual tokens, which becomes especially challenging for video generation tasks requiring high fidelity and temporal consistency.
>
> - **MLLM Freezing Strategy:** We keep the MLLM fully frozen to preserve its reasoning and alignment abilities. In contrast, methods like MetaMorph and JanusPro [3] fine-tune the MLLM to enable token generation, risking degraded performance without massive, high-quality data. As shown in Table 3 of the JanusPro paper, their multimodal understanding performance is much lower than that of MetaQuery and BLIP3o, both of which keep the backbone frozen. This supports our design choice to freeze the MLLM.
>
> - **Performance & Reproducibility:** In addition to the attached code in the supplementary material, we have made our model design, training data, and code fully transparent, and will open-source all components upon acceptance. In contrast, MetaMorph is still under legal review, has not released model weights, and lacks evaluation results on standard benchmarks like GenEval or DPGBench, making direct comparisons difficult. We also highlight that our method achieves performance close to SoTA models like MetaQuery and BLIP3o, both generally stronger baselines than MetaMorph, while requiring much lower training cost.
>
> Overall, we have done our best to objectively clarify the differences between our method and MetaMorph, providing detailed explanations and additional experiments to support our claims. We also highly value the contributions of MetaMorph. We hope this explanation helps illustrate the key technical and practical advantages of our method. If the reviewer has any further questions or suggestions, we would be more than happy to address them. Thanks again!
>
> [1] Autoregressive image generation without vector quantization, NeurIPS, 2024
>
> [2] Maskgit: Masked generative image transformer, CVPR, 2022
>
> [3] Janus-pro: Unified multimodal understanding and generation with data and model scaling, arXiv, 2025
>
> ---
>
> > ### W2: Additional quantitative scale-up results
>
> Beyond the results presented in the appendix, we obtained additional GPU resources after the submission deadline and conducted larger-scale experiments using the same training dataset as BLIP3-o [4], which is a recent SoTA method that connects MLLMs with diffusion models for image generation (released on May 14, 2025). Specifically, we use 27 million image-text pairs from the BLIP3o-Long-Caption dataset and train our model for 1.7 epochs, followed by fine-tuning on 60K high-quality images from the BLIP3o-60K dataset. In contrast, BLIP3-o is trained on the same 27M dataset for over 10 epochs. A comparison is summarized in the table below:
>
> | Models | MLLM Backbone | Total Pretraining Images (Training Steps×Batch Size) | GenEval | DPGBench |
> | :---- | :---- | :---- | :---- | :---- |
> | BLIP3-o | Qwen2.5-VL 3B | \~270M | 81% | 79.36% |
> | BLIP3-o | Qwen2.5-VL 7B | \~270M | 84% | 81.60% |
> | Bifrost-1 (Ours) | Qwen2.5-VL 3B | 9M | 61% | 76.41% |
> | Bifrost-1 (Ours) | Qwen2.5-VL 7B | 46M | 81% | 77.67% |
>
> As shown, the GenEval score improves from 61% to 81%, and DPGBench increases from 76.41% to 77.67%, demonstrating that better annotations and scaled-up training lead to meaningful gains. Notably, even with this larger-scale setup, our training cost remains much lower than prior works, while still outperforming existing approaches under the same compute budget.
>
> [4] Blip3-o: A family of fully open unified multimodal models—architecture, training and dataset, arXiv, 2025.
>
> ---
>
> > ### Q2: Paper restructure suggestion
>
> Thank you very much for the suggestion. We will move the generation benchmark table from the appendix, along with the new scaled-up experiment results, into the main paper to better emphasize their importance in the camera-ready version.
>
> ---
>
> Dear Reviewer 98F3,
>
> Once again, we sincerely thank you for all the comments and valuable feedback. We believe that we have addressed all the questions in depth. If there’s any additional questions, please feel free to let us know, and we would be more than happy to address them.

---

> > ### Comment · Reviewer_98F3 · 2025-08-03
> >
> > Thanks for the authors' rebuttal. The explanations of the contributions of this method and its conceptual overlap with MetaMorph, besides the new results, are not convincing to me.
> >
> > The ControlNet or Cross-attention control is not the contribution of this paper. Using different kinds of controls is only an experimental setting that is isolated from the claimed core contribution of "Patch-level CLIP Latents" which is the same as MetaMorph.
> >
> > The masked token decoding is also borrowed from MaskGIT and MAR.
> >
> > Using MLLM-aligned encoder is a natural experimental choice since the proposed method is based on a vision language model.
> >
> > Therefore, the rebuttal does not provide a clear distinction between its core contribution, "Patch-level CLIP Latents" and the MetaMorph.
> >
> > With a larger training dataset, the rebuttal provides better results on GenEval and DPG-Bench. However, on DPG-Bench, the method does not improve much and is still far behind most of the methods in Appendix Table 1. Since DPG-Bench contains more complex prompts than GenEval, I think DPG-Bench's results are more convincing to validate the models' unified understanding and generation. Therefore, the method does not get satisfactory improvement even with a larger training dataset.
> > Due to the above reasons, I retain my original score. I also encourage other reviewers and AC to investigate this paper more carefully and evaluate it more properly.

---

> > > ### Author Response · Authors · 2025-08-04
> > >
> > > Dear Reviewer 98F3,
> > >
> > > >### Clarifying our contributions: more than just patch-level CLIP latents
> > >
> > > We sincerely thank the reviewer for their thoughtful feedback on this point. We would like to take this opportunity to clarify that patch-level clip latents are only one design component of our model, instead of the sole contribution.
> > >
> > > The core motivation of our work stems from a broader question: **VLMs have already seen lots of images and videos through its training. Can VLMs assist image generation not by directly predicting precise visual tokens, but by producing coarse “image sketches” that already encode sufficient structure, layout, color, and texture?** We posit that such blurry or abstract visual representations may be more effective and efficient priors for guiding diffusion models than traditional cross-attention-based token conditioning.
> > >
> > > This insight leads to our design:
> > >
> > > - The choice of VLM (instead of LLM) as a backbone for a multimodal generation model is the key to our framework. We want to teach a VLM to learn to generate/output its visual representation, for which the model is already good at encoding. The use of patch-level clip latent was the natural choice because it is the same as its input, not because it was used by metamorph.
> > >
> > > - To make the 2D visual tokens process efficient and tightly coupled with the VLM’s knowledge, we introduce a trainable visual branch initialized from a copy of the VLM’s own visual parameters. This ensures strong native alignment between the VLM’s representations and the trainable modules.
> > >
> > > - The 2D visual tokens generated by VLMs serve as image sketches. This prior knowledge is then injected into the diffusion model to guide the final image generation through a ControlNet-based framework.
> > >
> > > - To further reduce training cost, we adopt Masked Autoregression (MAR) as an alternative to the traditional autoregressive decoding used in works like MetaMorph.
> > >
> > > From our experiments, we observe that these sketch-like 2D representations offer more effective conditioning than the image tokens typically used in cross-attention layers. Here's some intuitive explanations:
> > >
> > > - Cross-attention treats each image token as a global semantic feature, attending uniformly across space. For example, in DreamLLM (Figure 6), each query token contributes only partial semantic hints to the overall generation.
> > >
> > > - In contrast, our sketch-based tokens are spatially aligned with the 2D image latent space, with each token corresponding to a specific region or patch in the image, enabling more localized and structured conditioning.
> > >
> > > In summary, our paper introduces a new perspective on how VLMs can guide image generation by leveraging their learned visual knowledge to generate spatially grounded image sketches. And from the ablation experiments, we can see clearly that these sketches serve as a powerful and efficient prior for downstream generative models. Therefore, we would like to kindly disagree with the reviewer’s claim that our paper significantly overlaps with MetaMorph, and patch-level CLIP latents is just one of the design choices in our framework that can improve feature alignment and training efficiency.
> > >
> > > > ### Results on larger-scale experiments
> > >
> > > We acknowledge that our current performance on DPGBench is not yet as strong as some leading baselines (e.g., Janus Pro). However, we would like to emphasize that, as an academic lab, we do not have access to the same level of resources (e.g., high-quality proprietary data and large-scale compute) as major industry labs. Therefore, rather than focusing solely on benchmark scores, our primary goal is to contribute novel ideas to the community that demonstrate improved efficiency over existing methods through controlled and principled experimentation.
> > >
> > > To validate the effectiveness of our approach, we first conduct thorough evaluations on the ImageNet dataset, including comprehensive ablations and additional experiments presented in this rebuttal. We also report results on standard benchmarks such as GenEval and DPGBench to facilitate fair comparisons.
> > >
> > > We kindly highlight that the final benchmark scores are influenced by many factors, including training and fine-tuning datasets, compute budgets, and prompt preprocessing strategies—particularly whether prompts are rewritten using LLMs (e.g., MetaQuery, BAGEL). In our experiments, we strictly use the original, unmodified prompts provided in GenEval and DPGBench, without applying any LLM-based rewriting or simplification, in order to ensure fairness and transparency. Despite this, our GenEval results are already on par with strong baselines. Meanwhile, performance on DPGBench (whose prompts tend to be longer and more complex) suggests that additional training epochs may be necessary for the model to fully capture the intended semantics. We believe that with comparable training resources, our framework has potential to achieve further improvement.

---

> > > > ### Comment · Reviewer_98F3 · 2025-08-04
> > > >
> > > > Thanks for the authors' new response and clarifications.
> > > >
> > > > After carefully reading the response, I still find that the claimed contributions do not sufficiently distinguish the work from existing methods. Here are my reasons:
> > > >
> > > > 1.  **The choice of VLM (instead of LLM).**  The authors emphasize that selecting a VLM for multimodal generation is a key design choice. However, this idea is not novel. Prior work such as Kosmos-G (2023) has already explored leveraging VLMs/MLLMs for generation by exploiting their strong visual understanding. MetaMorph chose an LLM because it aimed to explore the mutual benefits between the training of both understanding and generation tasks. Therefore, choosing a VLM is not clearly a new conceptual contribution.
> > > >
> > > > Reference: Kosmos-G: Generating Images in Context with Multimodal Large Language Models
> > > >
> > > > 2.  **Control-Net or Cross-attention.** The distinction between using ControlNet conditioning or cross-attention conditioning appears to be an empirical design choice rather than a fundamentally new contribution. This is a useful experimental finding, but it is more of **an application of existing methods** than a novel contribution.
> > > >
> > > > 3. **Masked Autoregression (MAR).** The method adopts MAR for decoding, but MAR itself is a known method. It is again **an adaptation** rather than an innovation.
> > > >
> > > > 4. **Patch-level CLIP latents.** The main paper gives a lot of emphasis on this design, in the title, abstract, and main text. However, the response indicates that patch-level CLIP latents are used because they naturally match the VLM input. This further weakens the claim that this design provides a strong or unique contribution in the main paper.
> > > >
> > > > Therefore, the response does not sufficiently demonstrate clear novelty beyond prior works. The claimed contributions appear to be the **application** of existing components. I encourage the authors to discover clearer contributions from their designs.

---

> > > > > ### Author Response · Authors · 2025-08-04
> > > > >
> > > > > Dear Reviewer 98F3,
> > > > >
> > > > > Thank you again for your detailed comments. We would like to take this opportunity to clarify again that our choice of using ControlNet and cross-attention is not merely an empirical one. As we explained in our previous response, this design choice stems from our desire to explicitly guide the VLM to generate 2D image sketches, rather than just semantic tokens.
> > > > > With ControlNet, if the VLM doesn't generate meaningful 2D sketches, the image generation model won't benefit, because those sketches are simply added to the noisy latents elementwise. In contrast, cross-attention does not impose such spatial constraints, and the model can still transfer information without ensuring strict spatial alignment.
> > > > > Therefore, we would like to respectfully emphasize that our use of ControlNet is driven by the **conceptual alignment it offers with our design goals**, not simply because it performs better than cross-attention in empirical applications.
> > > > >
> > > > > ---
> > > > >
> > > > > In addition, we would like to point the NeurIPS 2025 reviewer guideline --- **“Originality does not necessarily require introducing an entirely new method. Rather, a work that provides novel insights by evaluating existing methods, or demonstrates improved efficiency, fairness, etc. is also equally valuable.”**
> > > > >
> > > > > Our method is not merely a combination of existing components. Instead, we carefully choose and integrate different design elements (VLMs vs. LLMs, ControlNet vs. cross-attention, autoregressive vs. MAR, etc.) to effectively implement and support our core idea: enabling VLMs to assist image generation by producing coarse “image sketches” that encode essential structural, layout, color, and texture information.
> > > > >
> > > > > As mentioned in the reviewer guideline, the novelty of a paper should not be judged solely on whether it introduces an entirely new architectural component. For instance, MetaMorph adopts autoregressive modeling to generate text and visual tokens jointly, a strategy that has already been explored in previous works such as EMU3 [1], Transfusion [2], Janus [3], and Chameleon [4]. Similarly, its use of cross-attention, as pointed out by the reviewer, is also well-established. Nevertheless, MetaMorph provides valuable insights into how understanding and generation can enhance one another, which contributes to its significance. In the same vein, our paper leverages architecture choices to validate and support our core ideas, rather than focusing purely on empirical gains or application-driven motivation.
> > > > >
> > > > > [1] Wang, Xinlong, et al. "Emu3: Next-token prediction is all you need." arXiv preprint arXiv:2409.18869 (2024).
> > > > >
> > > > > [2] Zhou, Chunting, et al. "Transfusion: Predict the next token and diffuse images with one multi-modal model." arXiv preprint arXiv:2408.11039 (2024).
> > > > >
> > > > > [3] Wu, Chengyue, et al. "Janus: Decoupling visual encoding for unified multimodal understanding and generation." Proceedings of the Computer Vision and Pattern Recognition Conference. 2025.
> > > > >
> > > > > [4] Team, Chameleon. "Chameleon: Mixed-modal early-fusion foundation models." arXiv preprint arXiv:2405.09818 (2024).
> > > > >
> > > > > ---
> > > > >
> > > > > Once again, we sincerely thank the reviewer for all the comments, feedback, and engagement in this rebuttal discussion. We believe this discussion is valuable for clarifying our novelty and contributions, as well as improving our manuscript. If there are any additional questions, please feel free to let us know. We would be more than happy to address them further.

---

> > > > > > ### Comment · Reviewer_98F3 · 2025-08-05
> > > > > >
> > > > > > Thanks for the authors' additional response.
> > > > > >
> > > > > > After carefully considering the new response, I remain unconvinced that the work provides sufficient novelty, original insights, or improved performance. While the response cites the NeurIPS guideline that **Originality does not necessarily require introducing an entirely new method. Rather, a work that provides novel insights by evaluating existing methods, or demonstrates improved efficiency, fairness, etc. is also equally valuable.** , I think the submission still does not meet these criteria. My reasons are organized below in line with these guidelines.
> > > > > >
> > > > > > 1. **Primarily a combination of existing components without enough new insights.** The method mainly integrates existing techniques, such as the VLM, ControlNet-based conditioning, patch-level CLIP latents, and MAR, without introducing its own conceptual or empirical contributions.
> > > > > >
> > > > > >     **Use of MAR.** This is a direct replacement of standard autoregression. Its efficiency benefits have already been validated in prior works, and the paper provides no new insights, analyses, or further experiments that extend this understanding.
> > > > > >
> > > > > >     **Use of VLM instead of LLM.** This direction has been explored in prior methods such as Kosmos-G. The paper does not provide new evaluations or evidence beyond the findings of existing work.
> > > > > >
> > > > > >     **Patch-level CLIP latents.** Most of the paper’s figures, tables, and discussions focus on this aspect. However, this operation is already used in MetaMorph.
> > > > > >
> > > > > > 2. **Not demonstrating convincing improvement in performance or efficiency.** As a whole pipeline, the benchmark performance remains modest on common benchmarks such as DPGBench, which is insufficient to claim a meaningful contribution under the NeurIPS guidelines.
> > > > > >
> > > > > > 3. **Insufficient analysis of design choices.** The paper does not convincingly analyze how its combination of VLM, MAR, and ControlNet provides a new understanding or finding. The design decisions are described, but there is little empirical or analytical discussion and experiments beyond what prior works have already proposed. As mentioned above, most the analyses are about the Patch-level CLIP latents in the original paper.
> > > > > >
> > > > > > As for the MetaMorph, it does not claim its combination or use of existing methods as a novelty.  It mostly provides insights, evaluations, and observations of how the visual generation ability emerges and how the joint training of understanding and generation influences each other through experimental evidence and analysis. However, as I mentioned above, the proposed paper lacks these novel analyses and evidence.
> > > > > >
> > > > > > Overall, the paper largely claims contributions (in the paper and responses) from existing methods without providing enough of its own novel insights, **validated** analyses, or improved performance. I think substantial additional works are needed for this paper to meet the standards of a conference like NeurIPS. These works may include deeper analysis, stronger ablations, clearer experimental evidence, and **reorganization of its contributions**.

---

> > > > > > > ### Author Response · Authors · 2025-08-06
> > > > > > > **Response to comments by Reviewer 98F3 (part 1/2)**
> > > > > > >
> > > > > > > Dear Reviewer 98F3,
> > > > > > >
> > > > > > > Thank you again for your continued engagement and thoughtful feedback. We truly appreciate the detailed discussion, and we understand your concerns regarding the originality and contributions of our work. We respectfully address your main points below:
> > > > > > >
> > > > > > >
> > > > > > > > ### Our new perspective on VLM and integration strategy of different building blocks
> > > > > > >
> > > > > > > We agree that our method builds upon existing components like ControlNet, patch-level CLIP latents, and MAR. However, as the NeurIPS reviewer guidelines note, originality also includes "novel insights by evaluating existing methods" and demonstrating “improved efficiency, fairness, etc.”
> > > > > > >
> > > > > > > Our core contribution is not the invention of a new component in isolation, but rather a **new perspective and integration strategy**: we demonstrate that a pretrained VLM can serve as a spatial prior generator for image synthesis by outputting coarse 2D visual sketches, rather than directly generating final images or discrete tokens. This insight motivates our architectural decisions (e.g., ControlNet-based latent injection, VLM-initialized trainable vision branches, frozen MLLM backbone) and leads to several practical advantages in efficiency and alignment, as validated in our experiments.
> > > > > > >
> > > > > > > In addition, we would like to kindly reminder the reviewer that **The novelty of our idea, supported by solid experiments, is also well acknowledged by the other reviewers:**
> > > > > > >
> > > > > > > - Reviewer NRRC:
> > > > > > >   - “Novel and Clever Bridging Mechanism: The central idea of using natively-aligned, patch-level CLIP latents as the bridge is novel and elegant. It sidesteps the more complex alignment challenges faced by other methods and provides a conceptually simple way to connect the understanding capabilities of MLLMs with the synthesis power of diffusion models.”,
> > > > > > >   - “Preservation of MLLM Capabilities: A key advantage is that the MLLM's original parameters are untouched, ensuring its strong multimodal understanding and reasoning abilities are fully preserved.”
> > > > > > >
> > > > > > > - Reviewer k2H1:
> > > > > > >   - “The overall performance is solid, and the experimental setup is well-structured. The experiments convincingly support the effectiveness of the proposed method.”
> > > > > > >   - “Extensive ablation studies and benchmark evaluations clearly demonstrate the contributions of the proposed modules.”
> > > > > > >   - “The design choices are well justified through ablation studies, including comparisons between CLIP latents and other alternatives like VAE or learnable 2D query tokens.”
> > > > > > >
> > > > > > > - Reviewer XePf:
> > > > > > >   - “The experiments show that the proposed architecture injects image generation capabilities with a relatively small training cost while maintaining the MLLM inference ability.”
> > > > > > >
> > > > > > > ---
> > > > > > >
> > > > > > > > ### Use of MAR
> > > > > > >
> > > > > > > The use of MAR instead of standard autoregression is a natural choice in our paper, as our goal is not to design a unified autoregressive framework for multimodal understanding and generation, but rather to extract meaningful 2D image sketches from VLMs for downstream image generation. This choice aligns well with our pipeline and brings improved training efficiency, as clearly demonstrated in previous papers [1, 2] and acknowledged by the reviewer: “Its efficiency benefits have already been validated in prior works.”
> > > > > > >
> > > > > > > Furthermore, we would like to emphasize that **prior works employing the MAR strategy, such as Show-o [3] and NOVA [4], also did not re-run comparisons between AR and MAR, yet were still well-received at top conferences (e.g., ICLR’25)**. We believe that building upon established methods and findings from prior studies should not be viewed as a limitation.
> > > > > > >
> > > > > > > Importantly, **we do not claim that the use of MAR is a novel contribution to our paper.** Investigating new variants or findings related to MAR is orthogonal to our main goal, which is to leverage VLMs to enhance image generation through 2D spatial visual priors
> > > > > > >
> > > > > > > [1] Li, Tianhong, et al. "Autoregressive image generation without vector quantization." Advances in Neural Information Processing Systems 37 (2024): 56424-56445.
> > > > > > >
> > > > > > > [2] Fan, Lijie, et al. "Fluid: Scaling autoregressive text-to-image generative models with continuous tokens." arXiv preprint arXiv:2410.13863 (2024).
> > > > > > >
> > > > > > > [3] Xie, Jinheng, et al. "Show-o: One single transformer to unify multimodal understanding and generation." arXiv preprint arXiv:2408.12528 (2024).
> > > > > > >
> > > > > > > [4] Deng, Haoge, et al. "Autoregressive video generation without vector quantization." arXiv preprint arXiv:2412.14169 (2024).

---

> > > > > > > ### Author Response · Authors · 2025-08-06
> > > > > > > **Response to comments by Reviewer 98F3 (part 2/2)**
> > > > > > >
> > > > > > > > ### Choice of VLM instead of LLM
> > > > > > >
> > > > > > > We would like to respectfully remind the reviewer that we did **not** claim to be the first to explore the use of VLMs for image generation. We fully acknowledge that many prior works, such as Kosmos-G, have investigated this direction, and we have clearly cited these works in the related work section (L81–L102).
> > > > > > >
> > > > > > > However, unlike those prior studies that primarily use VLMs to condition image generation on textual or contextual information, our work is, to the best of our knowledge, **the first** to explore how a VLM can generate implicit 2D spatial visual priors for image generation while keeping all parameters in the image diffusion model fully frozen.
> > > > > > >
> > > > > > > **We choose to use a VLM rather than an LLM for a straightforward reason: LLMs do not come with natively aligned visual encoders.** As shown in the quantitative results from our previous response, using natively aligned visual encoders leads to clear performance improvements over external encoders like SigLIP.
> > > > > > >
> > > > > > > Therefore, the reviewer's statement that "The paper does not provide new evaluations or evidence beyond the findings of existing work." appears to overlook both the evaluation results presented in the paper (e.g., Table 1) and the additional findings shared in our previous responses.
> > > > > > >
> > > > > > > ---
> > > > > > >
> > > > > > >
> > > > > > > > ### Use of patch-level CLIP latents
> > > > > > >
> > > > > > > We kindly remind the reviewer that we have already clearly discussed our differences with MetaMorph in our previous response, **“Clarifying our contributions: more than just patch-level CLIP latents.”** It is unfair to repeat the same claim without considering our earlier clarifications.
> > > > > > >
> > > > > > >
> > > > > > > ---
> > > > > > >
> > > > > > > > ### Improvement in performance or efficiency
> > > > > > >
> > > > > > > We kindly remind the reviewer that we have already discussed our large-scale experimental results in detail in our previous response under **“Results on larger-scale experiments.”** In short, even without access to high-quality internal proprietary data or large-scale GPU resources, our method achieves performance that is nearly on par with strong existing baselines such as BLIP3-o (which, released on May 14, 2025, should be considered a contemporary work according to the NeurIPS guidelines), while requiring significantly less training compute.
> > > > > > >
> > > > > > > Additionally, **as shown in Table 1 of our paper and in the new results provided during the rebuttal, our method demonstrates much better training efficiency compared to several alternative design choices**. These include using non-MLLM-aligned visual encoders (e.g., VAE or SigLIP), latent ControlNet over cross-attention, and achieving robust image generation quality while reducing MLLM decoding steps by up to 8 times. Therefore, the reviewer’s statement that we are “not demonstrating convincing improvement in efficiency” is unfounded.
> > > > > > >
> > > > > > > ---
> > > > > > >
> > > > > > > > ### Analysis of design choices
> > > > > > >
> > > > > > > The reviewer’s statement that “The design decisions are described, but there is little empirical or analytical discussion and experiments beyond what prior works have already proposed” clearly overlooks the experiments presented in our paper, as well as the new results provided during the rebuttal period. **We have included several experiments that, to the best of our knowledge, have not been shown in prior works and that directly support our design choices**, including:
> > > > > > >
> > > > > > > - A comparison between MLLM-aligned visual encoders, non-MLLM-aligned encoders (e.g., VAE or SigLIP), and learnable query tokens
> > > > > > >
> > > > > > >
> > > > > > > - An analysis comparing our latent ControlNet to cross-attention for the specific task of injecting 2D image priors
> > > > > > >
> > > > > > >
> > > > > > > - An ablation study showing that the model remains robust even when reducing MLLM decoding steps by up to 8×
> > > > > > >
> > > > > > >
> > > > > > > We also include additional experiments to support our paper, such as the scalability of the number of VLM-generated image tokens and large-scale evaluations on GenEval and DPGBench. We believe it is unfair for the reviewer to make such a claim without pointing to any specific missing or critical experiments that would be necessary to support our design choices.
> > > > > > >
> > > > > > > ---
> > > > > > >
> > > > > > > While we are happy to reorganize our contributions as the reviewer suggested, we respectfully point out that several statements are already addressed in our previous responses. We kindly encourage the reviewer to reevaluate our work.

---

> ### Comment · Reviewer_98F3 · 2025-08-06
>
> Thank you to the authors for the new response. I would like to clarify my position and provide my updated comments below.
>
> >1. **Clarification of my previous comments.**
>
> When I stated that “Overall, the paper largely claims contributions from existing methods”, I was referring to both the paper and the **responses**. In the authors' earlier response ("Clarifying our contributions: more than just patch-level CLIP latents"), the MAR is explicitly listed. This is why I interpreted it as being claimed as a contribution or a difference from other methods.
>
> >2. **New experimental results.**
>
> For Table 2 in the response, my understanding is that the new experiments replace the VLM's original visual encoder with SigLIP or VAE.  It is surely natural that these two variants perform worse when trained with only ImageNet with 7 epochs for the MLLM vision branch and 1 epoch for ControlNet. This is because the features are not aligned, and the LLM cannot understand the new image features. While these experiments show a performance gap, I do not find that they provide new insights. The authors' explanation, "We choose to use a VLM rather than an LLM for a straightforward reason: **LLMs do not come with natively aligned visual encoders.**" is already well known and has been explicitly discussed in Kosmos-G.
> Kosmos-G highlights that using MLLMs for generation benefits from:
> 1) It capitalizes on the **inherent vision-language alignment within the MLLM**.
> 2) The MLLM architecture naturally supports interleaved interleaved multi-image and text input.
> 3) The pre-trained MLLM can effectively model multimodal input in context.
>
> These reasons appear very similar to the authors' current explanation.
>
> That is also why MetaMorph needs to perform training of vision understanding jointly with the generation. This paper says, "**(1) visual generation ability emerges as a natural byproduct of improved visual understanding, and can be unlocked efficiently with a small amount of generation data;"**. Therefore, I am not convinced that this new evaluation offers fundamentally new insights.
>
> >3. **Repetition of my concern about claimed contributions.**
>
> I repeat the same claim of "Clarifying our contributions: more than just patch-level CLIP latents." because I remain unconvinced by the new claims in this section, including ControlNet-conditioning, VLM, and MAR, etc. My concerns are stated in this and previous responses.
>
> >4. **MAR and prior works.**
>
> I would like to clarify that I don't quite agree that "that prior works employing the MAR strategy, such as Show-o and NOVA, also did not re-run comparisons between AR and MAR, yet were still well-received at top conferences (e.g., ICLR’25). We believe that building upon established methods and findings from prior studies should not be viewed as a limitation.". Citing that prior works such as Show-o were accepted without re-running AR vs. MAR comparisons does not address the core issues:
>
> 1. My concern is **always not about using them**. It is about **claim them as a difference with other methods**. In the authors' response: "Clarifying our contributions: more than just patch-level CLIP latents.", the MAR is listed, and this is why I think it is claimed as a difference or a contribution.
>
> 2. The acceptance of previous papers is not a sufficient reason to justify this paper's contribution under NeurIPS standards.
>
> >5. **The performance and comparison on DPGBenceh.**
>
> Results that rely on untested experiments due to GPU limitations should remain **unknown**, and should not be implicitly interpreted as supporting evidence for efficiency or performance claims. From the current results and my above analysis, I am not convinced that this paper gives a notable improvement.
>
> >6. **Reviewing fairness.**
>
> I want to emphasize that my continued discussion is not because I want to be an **unfair** reviewer. This is because I am not convinced and would like to make my concerns and questions as clear as possible. I think this is normal in a reviewing process. I am also confident that I have a deep understanding of this area, which is why my review and responses are **much detailed than other reviewers**, and I aim to ensure my response is both clear, reasonable, and **well-supported by related works**.

---

> ### Author Response · Authors · 2025-08-06
>
> We sincerely thank the reviewer for the feedback. The following are our responses to the reviewer's corresponding questions:
>
> > ### Clarification of the reviewer’s previous comments
>
> First, we would like to kindly clarify that our paper does not claim MAR as a novelty or contribution (see L72–L79). In our response, we mentioned MAR only because **the reviewer explicitly asked us to clarify the significant differences between our method and MetaMorph**. In doing so, we outlined our high-level motivation and corresponding design differences. We explicitly stated that MAR is not a novel idea proposed by us, nor is it a primary contribution of our work. Rather, we use it to further reduce training cost.
>
> > ### New experimental results
>
> Regarding this second point, we never claimed that using an MLLM to assist image generation is our novelty. As clearly stated in our response, **our contribution lies in using an MLLM to generate implicit 2D spatial image priors to guide downstream image generation**. To the best of our knowledge, this specific design has not been explored in previous works.
>
> As supported by our empirical results, we show that spatial tokens injected via ControlNet, added element-wise to the diffusion latents, offer more effective spatial conditioning than the cross-attention mechanism used in MetaMorph. In this sense, our approach **complements** the findings in MetaMorph (which suggest that strong understanding capabilities can enhance generation), and can be applied as a more effective model architecture compared with the cross-attention used in MetaMorph.
>
> We believe our work offers a valuable new perspective: that VLM-generated image tokens can serve as effective priors for guiding diffusion models. We respectfully suggest that the reviewer may have misunderstood our core idea, which was clearly recognized by other reviewers as noted in our earlier responses.
>
> > ### Repetition of the reviewer’s concern about claimed contributions
>
> As for the third point, we believe our previous response has already answered the reviewer’s question. If there are specific additional experiments that the reviewer believes essential to support our core claims, we are more than happy to conduct them and discuss further.
>
> > ### MAR and prior works
>
> Regarding our response to the reviewer’s original question about differences from MetaMorph, we did our best to answer thoroughly, including modeling differences such as AR vs. MAR. Highlighting such differences is a direct response to the reviewer’s request, and we do not believe doing so should be considered a weakness.
>
> > ### The performance and comparison on DPGBenceh
>
> For efficiency, we presented two sets of experiments.
>
> - First, in Table 1 and additional rebuttal experiments on ImageNet, we compared various architectural designs under the same data and compute budgets. **These results demonstrate that our architecture is more efficient than alternatives under a fair and strictly controlled setting**.
>
> - Second, on GenEval and DPGBench, a fair apples-to-apples comparison is challenging due to the varying training data, compute resources, and model designs used by different baseline methods. Therefore, our experiments in the paper along with the additional scaling experiments, are intended to demonstrate that our method achieves performance almost comparable to existing baselines while using significantly fewer resources.
>
> > ### Reviewing fairness
>
> Lastly, we would like to clarify that we never intended to suggest the reviewer was being unfair. Our comments referred solely to specific statements that we believe may not fully align with the content of our paper or our responses. If our wording gave the wrong impression, we sincerely apologize. We have great respect for all reviewers and deeply value the peer review process.
>
>
> ---
>
> ### Final Notes
>
> We thank the reviewer for their continued feedback and greatly appreciate the time and effort spent engaging with our work.
> As our shared goal is to explore open research questions and contribute good ideas to the community, we remain happy to provide further clarifications or conduct additional experiments if needed. While we appreciate the reviewer’s perspective and will keep their suggestions in mind for future extensions, we believe that the current version, together with the additional experiments included during the rebuttal, already provides strong support for our claims.

---

### Official Review · Reviewer_NRRC · 2025-07-03

**Clarity:** 4
**Significance:** 3
**Originality:** 3
**Rating:** 5
**Confidence:** 5

**Summary:**

This paper proposes BIFROST-1, a framework to bridge pre-trained Multimodal Large Language Models (MLLMs) with diffusion models for high-fidelity image generation. The core idea is to use patch-level CLIP image embeddings, which are natively aligned with the MLLM's visual encoder, as the communication medium. The MLLM is trained with a lightweight visual generation branch to predict these embeddings, which then guide a pre-trained diffusion model via a lightweight latent ControlNet. This approach aims to achieve efficient training and strong generation performance while preserving the MLLM's original reasoning capabilities by keeping its backbone frozen.

**Questions:**

- Could you provide standard quantitative metrics (e.g., GenEval and DPGBench) comparing BIFROST-1 with other leading text-to-image generation models like JanusPro or MetaQuery? The qualitative results in Figure 7 are promising, but objective numbers are needed.
- Could you provide any quantitative results or qualitative examples demonstrating the model's ability to handle complex compositional prompts that require precise spatial reasoning and object attribute binding?
- What is the inference latency of your method, particularly the autoregressive step for generating patch embeddings? How does it compare to other methods that generate guidance signals in a single forward pass?

**Ethical Concerns:**

["NO or VERY MINOR ethics concerns only"]

**Final Justification:**

After reading the other reviewers' comments and the authors' response, I find that my initial concerns have been well addressed. Although reviewer 98F3 questions the novelty of using patch-level CLIP latents, I do not believe that is the core contribution of this paper. Instead, the key strength lies in how the authors effectively leverage patch-level CLIP latents to achieve structured guidance through a lightweight ControlNet. Overall, I am convinced by the authors’ clarification and experimental validation, and therefore decide to increase my score.

**Limitations:**

Yes

**Quality:**

4

**Strengths And Weaknesses:**

### Strengths
+ Novel and Clever Bridging Mechanism: The central idea of using natively-aligned, patch-level CLIP latents as the bridge is novel and elegant. It sidesteps the more complex alignment challenges faced by other methods and provides a conceptually simple way to connect the understanding capabilities of MLLMs with the synthesis power of diffusion models.
+ High Training Efficiency: By freezing the large MLLM and diffusion model backbones and only training lightweight components, the proposed method is highly computationally efficient. The ablation study in Table 1 effectively demonstrates the superiority of this design choice over alternatives.
+ Preservation of MLLM Capabilities: A key advantage is that the MLLM's original parameters are untouched, ensuring its strong multimodal understanding and reasoning abilities are fully preserved. This is well-supported by the benchmark results in Table 2.

### Weaknesses
- Insufficient Quantitative Comparison on Image Generation: The paper lacks a direct quantitative comparison against other state-of-the-art text-to-image generation models on standard benchmarks (e.g., DPG-Bench or GenEval). While Figure 7 provides qualitative examples, objective metrics like FID or CLIP Score are necessary to properly situate the model's performance. The Appendix is mentioned, but this critical comparison should be in the main paper for a thorough evaluation.
- Limited Evaluation of "Controllability": While the framework is presented as enabling "controllable" generation, the experiments primarily focus on image reconstruction and relatively simple prompts. There is a lack of validation on complex compositional prompts involving multiple objects with specific spatial relationships. This makes it difficult to assess the true extent of control the method provides.
- Unaddressed Inference Latency: The method relies on an autoregressive masking process to generate the patch embeddings during inference. This iterative process can be slow. The paper completely omits any analysis or reporting of inference speed, which is a significant oversight for a method aiming for practical utility.

---

> ### Author Rebuttal · Authors · 2025-07-31
>
> We would like to sincerely thank the Reviewer for the feedback. Since the weakness and questions of the Reviewer are closely related, we use “W/Q” to represent the answer to the corresponding weakness and questions.
>
> > ### W1/Q1: Standard quantitative metrics (e.g., GenEval and DPGBench) comparing BIFROST-1 with other leading T2I generation models.
>
> Thank you very much for the useful comment. We would like to first kindly point out that we have included standard metrics from GenEval and DPGBench in Appendix Table 1 (in the PDF of the supplementary materials, as mentioned in L284-285 of our main paper). In Appendix Table 1, we measure the training cost by the total number of training images passed to the model, calculated as the product of the number of training steps and the batch size per step. As we can see, our method achieves performance comparable to baselines such as Janus and EMU, while significantly lower training cost (training data 9M from Ours v.s. 100M from Janus).
>
> Beyond the results presented in the appendix, we gained more GPU support after the paper submission deadline, and here we include additional larger-scale experiments trained on the same training dataset from BLIP3-o [1], one of the latest works that connects MLLM with diffusion models for image generation released on May 14, 2025. Specifically, we use 27 million image-text pairs from BLIP3o-Long-Caption dataset, and train the model for 1.7 epochs, followed by fine-tuning on 60k high-quality images in the BLIP3o-60K dataset. In contrast, BLIP3-o is trained on the same 27M dataset for over 10 epochs. We summarize the comparison in the following table:
>
> | Models | MLLM Backbone | Total Pretraining Images (Training Steps×Batch Size) | GenEval | DPGBench |
> | :---- | :---- | :---- | :---- | :---- |
> | BLIP3-o | Qwen2.5-VL 3B | \~270M | 81% | 79.36% |
> | BLIP3-o | Qwen2.5-VL 7B | \~270M | 84% | 81.60% |
> | Bifrost-1 (Ours) | Qwen2.5-VL 3B | 9M | 61% | 76.41% |
> | Bifrost-1 (Ours) | Qwen2.5-VL 7B | 46M | 81% | 77.67% |
>
> As shown, the GenEval score improves significantly from 61% to 81%, and the DPGBench score increases from 76.41% to 77.67%. This demonstrates that higher-quality training data with better annotations, as well as scaled-up training, can lead to meaningful performance gains. Lastly, we would like to emphasize that even with this larger-scale training, our training cost remains much lower than that of prior works, and our method still outperforms previous approaches under the same training budget.
>
> [1] Chen, Jiuhai, et al. "Blip3-o: A family of fully open unified multimodal models—architecture, training and dataset." arXiv preprint arXiv:2505.09568 (2025).
>
>
> ---
>
> > ### W2/Q2: Quantitative results or qualitative examples demonstrating the model's ability to handle complex compositional prompts that require precise spatial reasoning and object attribute binding
>
> Thank you very much for this question regarding the model’s ability to handle complex prompts. First, we would like to apologize that, due to the NeurIPS rebuttal policy, we are not allowed to upload any qualitative examples or visualizations. However, we believe the benchmarks used in our evaluation on benchmarks including GenEval and DPGBench already include prompts that assess various capabilities such as object count, spatial and non-spatial relationships, colors, shapes, textures, and attribute binding between objects and their properties.
>
> We compared our method (Bifrost-1) with BLIP3-o on several out-of-distribution prompts involving rare color–attribute bindings, spatial relationships, and object counts. These include examples such as: “A photo of a blue pizza and a yellow baseball glove”, “A photo of four giraffes”, “A photo of a skateboard above a person”, “A purple computer keyboard and a red chair”. Our results demonstrate that Bifrost-1 generates more accurate and faithful images compared to BLIP3-o. We promise to include additional visualizations and comparisons with such complex prompts in our camera-ready version.
>
> Additionally, we would like to highlight that due to our design choice of keeping the MLLM fully frozen, the model is able to preserve its strong reasoning capabilities for understanding complex prompts. For example, given a prompt requiring implicit reasoning such as “The national flag of the country where Yellowstone National Park is located,” the MLLM can infer that the intended flag is that of the United States. It can then rewrite the prompt as “The national flag of the United States” before passing it to the visual generation branch. This ability to transform complex or implicit prompts into explicit and visual-friendly ones greatly enhances the model’s overall effectiveness in handling sophisticated user instructions.
>
> ---
>
> > ### W3/Q3: Inference speed for autoregressive masking process
>
> Thank you very much for raising this valuable point regarding the inference speed of our model, particularly the autoregressive masking process. We agree that inference speed is an important consideration for practical applications, and that balancing efficiency and accuracy is a critical design trade-off. Below, we first outline the details of our inference procedure and then present new quantitative results using different numbers of decoding steps in the MLLM.
>
> - **Detailed inference procedure:** During inference, we begin with a fully masked 2D token grid, which is fed into the MLLM’s vision generation branch. The accompanying text condition is provided to the frozen MLLM branch to guide the visual token prediction. Prior to decoding, we randomly sample a generation order over the image patch tokens, which determines the sequence in which masked tokens are decoded into clean ones. Following MaskGIT [2], we adopt a cosine schedule to determine how many tokens are unmasked at each decoding step. This schedule, as visualized in Figure 2 of the MaskGIT paper, allocates fewer unmasked tokens in the early stages (when uncertainty is high), and progressively more in later steps as more tokens become available. In our default implementation, we set the number of decoding steps equal to the number of visual tokens for simplicity.
>
> - **Single step decoding v.s. multistep decoding:** In Table 1 of our main paper, we present an ablation study in which patch-level CLIP latents are replaced by 2D learnable query tokens, predicted in a single forward pass. This setup effectively performs a single-step autoregressive prediction and leads to significantly worse generation quality, with an FID of 118.69, compared to our default approach which achieves 25.77.
>
> - **Quantitative Results with Varying Decoding Steps:** To further assess the effect of decoding steps, we conducted experiments with varying numbers of decoding passes on ImageNet. The results, summarized in the table below, show image quality metrics including FID, sFID, and IS. As demonstrated in the table below, our method remains robust as long as the number of decoding steps is greater than 8. There is, however, a clear trade-off between inference speed and image quality when using fewer decoding steps. Importantly, we also highlight that the MLLM decoding time is significantly smaller than the runtime of the diffusion-based image generation model. For instance, FLUX, a 12B parameter model, takes **14.79** seconds to generate a single image with the default 28 denoising steps. Therefore, the MLLM decoding time is not a major bottleneck. From a practical standpoint, users can flexibly select the number of decoding steps based on their specific priorities—whether they favor faster inference or higher image quality for their particular application.
>
> [2] Chang, Huiwen, et al. "Maskgit: Masked generative image transformer." Proceedings of the IEEE/CVF conference on computer vision and pattern recognition. 2022.
>
> | Number of decoding steps | 64 (default) | 32 | 16 | 8 | 4 | 2 | 1 |
> | :---- | :---- | :---- | :---- | :---- | :---- | :---- | :---- |
> | Clock time | 5.21s | 2.63s | 1.37s | 0.66s | 0.35s | 0.19s | 0.09s |
> | FID (↓) | 18.64 | **18.49** | 18.79 | 18.89 | 19.97 | 24.90 | 60.35 |
> | sFID (↓) | 43.64 | **43.58** | 44.52 | 45.89 | 50.86 | 67.23 | 217.50 |
> | IS (↑) | 156.01 | **158.93** | 156.96 | 158.87 | 150.56 | 124.78 | 83.22 |
>
> ---
> Dear Reviewer NRRC,
>
> Once again, we sincerely thank you for all the comments and valuable feedback. We believe that we have addressed all the questions in depth, in particular: (1) additional experimental results comparing Bifrost-1 with other T2I models on standard quantitative metrics, (2) evaluation of the model’s ability to handle complex prompts beyond GenEval and DPGBench, and (3) inference speed of the autoregressive masking process, including experiments with fewer decoding steps. If the reviewer has any additional questions, please feel free to let us know, and we would be more than happy to address them.

---

> ### Author Response · Authors · 2025-08-08
> **Sincere thank for the Reviewer's feedback**
>
> Dear Reviewer NRRC,
>
> As today marks the end of the author-reviewer discussion period, we would like to express our heartfelt thanks for your valuable feedback and thoughtful suggestions throughout the rebuttal process.
>
> We sincerely appreciate your recognition of our work, particularly the novel and elegant bridging mechanism using patch-level CLIP latents to connect MLLMs and diffusion models, the training efficiency achieved by freezing the main backbones and updating only lightweight modules, and the preservation of the MLLM’s original understanding and reasoning capabilities.
>
> If you have any remaining questions or concerns, please don’t hesitate to let us know, and we would be more than happy to address them by the end of today.
>
> Thank you again for your time and effort!

---

### Note · Authors · 2025-08-15

Dear AC,

We sincerely thank you and the reviewers for your time and feedback. Below, we summarize the strengths recognized by the reviewers and responses to their questions.

---
> ### Strengths

- **Novelty & Design**: NRRC, k2H1, and XePf praised the novelty of using natively aligned, patch-level CLIP latents as a simple yet powerful bridge between MLLM understanding and diffusion synthesis, sidestepping complex alignment challenges faced by prior methods.
- **Preservation of MLLM’s Reasoning Capabilities**: As noted by NRRC, freezing the MLLM’s original parameters ensures its multimodal understanding abilities remain intact.
- **Empirical Usefulness with Extensive Experiments**: k2H1 and XePf commended our solid performance, well-structured experiments, extensive ablations, benchmark evaluations, and the low cost of adding image generation capability.

---
> ### Summary of Responses to Reviewers’ Concerns

- **Common concern**: Added standard quantitative results (GenEval, DPGBench) with larger-scale training, supplementing Appendix Table 1.
- **NRRC’s concern**: Ability to handle complex compositional prompts and inference efficiency.
  - We clarified the model’s handling of such prompts and quantified inference speed for the autoregressive masking process under varying decoding steps.
- **k2H1’s concern**: Justification for latent ControlNet and architectural choices.
  - Explained our choice over pretrained ControlNets, provided a detailed architectural description, and analyzed generation fidelity and diversity as CLIP token increases.
- **XePf’s concern**: Differences from related methods and clarity of results.
  - Clarified differences with the method in Fig. 1(b) and our experiments in Sections 5.1–5.3.
- **98F3’s concern**: (1) Novelty vs. prior works; (2) Efficiency and performance.
  - **Difference with previous works:** Highlighted using an MLLM to generate implicit 2D spatial image priors is an unexplored direction. Described technical distinctions from MetaMorph and KosmosG.
  - **Training efficiency and performance:** Presented results (Table 1, Appendix Table 1, and new rebuttal experiments) showing our architecture is more efficient than alternatives under fair, controlled settings, while achieving competitive performance with fewer resources.

---
We are glad most reviewers clearly recognized the novelty, contributions, and empirical strengths of our work. We will incorporate the clarifications and experiments into the final version.

---

### Decision · Program_Chairs · 2025-09-17

**Decision:**

Accept (poster)

**Comment:**

The paper presents a framework that bridges MLLMs and diffusion models using patch-level CLIP latents, a lightweight autoregressive visual branch, and a Latent ControlNet. By freezing the MLLM backbone and training only small auxiliary modules, the approach preserves reasoning ability while enabling efficient and competitive image generation. Experiments show favorable trade-offs among fidelity, reasoning retention, and compute efficiency.

The AC has carefully read the paper, the reviews, the rebuttal, and all post-rebuttal discussions. The AC also appreciates the high-quality, in-depth exchange between the reviewers and authors, which clarified key issues around novelty, positioning, and empirical validation. Reviewer 98F3 raised important concerns about conceptual overlap with prior efforts (e.g., MetaMorph) and the reliance on existing components such as CLIP and ControlNet. The AC highly values this perspective, but after weighing the rebuttal and subsequent discussion, the AC finds that the contribution is not merely incremental. While several modules build on known ideas, their use here is well-motivated toward enabling patch-level CLIP latents to serve as an effective bridge. Rather than being incidental add-ons, these designs are central to making the CLIP-latent interface practical and efficient, consistent with the core positioning of the work. From this perspective, the integration is more significant than the critique suggested. In addition, the other reviewers were overall supportive: they highlighted the practicality of using patch-level CLIP latents as a bridging mechanism, the elegance of freezing the backbone to preserve reasoning ability, the breadth of evaluation, and the strong efficiency benefits relative to comparable systems. Taking all perspectives together, the AC concludes that—even if individual pieces have precedents—the integrated framework is coherent, efficient, and valuable. The balance of evidence supports acceptance.

The AC also suggests that the authors acknowledge related efforts such as X-VILA, which similarly addresses visual information loss in cross-modality alignment; this is a useful connection but not a key weakness of the submission. For the camera-ready. Please (i) incorporate all new results and analyses introduced during rebuttal into the main paper with clear positioning; (ii) refine the novelty discussion and related-work section to clearly delineate differences from MetaMorph and adjacent bridging approaches, and add missing citations; (iii) consolidate ablations/design justifications for key components and report training/inference costs in a reproducible manner; and (iv) briefly discuss limitations and the most promising paths for improvement.